# `LDA2Net` Digging under the surface of *COVID-19 scientific literature* topics via a network-based approach

**Giorgia Minello**[1]*, **Carlo Romano Marcello Alessandro Santagiustina**[2,3], **Massimo Warglien**[2]

**1** Department of Environmental Sciences, Informatics and Statistics, Ca' Foscari University, Venice, Italy, **2** Department of Management, Ca' Foscari University, Venice, Italy, **3** Venice International University, Venice, Italy

☯ These authors contributed equally to this work.
* giorgia.minello@unive.it

**Data Availability Statement:** The COVID-19 Open Research Dataset (CORD-19) can be found at the following link: https://github.com/allenai/cord19.

**Funding:** The authors acknowledge financial support from the European Union Horizon 2020

## Abstract

During the COVID-19 pandemic, the scientific literature related to SARS-COV-2 has been growing dramatically. These literary items encompass a varied set of topics, ranging from vaccination to protective equipment efficacy as well as lockdown policy evaluations. As a result, the development of automatic methods that allow an in-depth exploration of this growing literature has become a relevant issue, both to identify the topical trends of COVID-related research and to zoom-in on its sub-themes. This work proposes a novel methodology, called `LDA2Net`, which combines topic modelling and network analysis, to investigate topics under their surface. More specifically, `LDA2Net` exploits the frequencies of consecutive words pairs (i.e. *bigram*) to build those network structures underlying the hidden topics extracted from large volumes of text by Latent Dirichlet Allocation (LDA). Results are promising and suggest that the topic model efficacy is magnified by the network-based representation. In particular, such enrichment is noticeable when it comes to displaying and exploring the topics at different levels of granularity.

## Introduction

The massive response of the scientific community to the COVID-19 pandemic has produced an unprecedented amount of research and related publications. For example, the Cord-19 open research dataset [1] currently includes over five hundred thousand published peer-reviewed and pre peer-reviewed articles. This tremendous volume of COVID-related works and the fast emergence of new research branches far exceeds the human capability to meaningfully organize and explore such material. This makes it impossible, even for a specialist, to map, explore and summarise such a massive corpus of documents without the help of automatic tools that can extract and classify useful semantic information from unstructured texts. A key issue related to analyzing large sets of text, such as scientific corpora, consists of identifying topics that may spread across several disciplines or different research branches. In this

project ISEED (Grant Agreement No. 960366). C.S. also acknowledges financial support from the European Union Horizon 2020 project MUHAI (Grant Agreement No. 951846) and from PON R&I 2014-2020 (FSE REACT-EU). G.M. also acknowledges financial support from project iNEST (Interconnected NordEst Innovation Ecosystem), funded by European Union Next - GenerationEU - National Recovery and Resilience Plan (NRRP) - MISSION 4 COMPONENT 2, INVESTMENT N. ECS00000043 - CUP N. H43C22000540006. In this regard, the manuscript reflects only the authors' views and opinions; neither the European Union nor the European Commission can be considered responsible for them.

**Competing interests:** The authors have declared that no competing interests exist.

framework, topic models, i.e. text processing techniques belonging to the family of statistical methods, are excellent tools to categorize documents and extract information from textual data, developed precisely for capturing subject matters occurring in a collection of documents.

Many works, such as [2], have attempted to classify and summarize the vast COVID-related literature collected in the Cord-19 dataset through topic modeling approaches. However, despite offering a categorization of documents, these analyses often remain at the surface level, particularly when exploring and understanding latent topics and their internal semantic structure. This drawback is mainly because most of these models are based on the so-called "bag-of-words" approach, which is unsuitable for analyzing texts semantically as it ignores the order and semantic relationship between words.

In this paper, we address the "bag-of-words" limit of the most popular topic modeling technique, the Latent Dirichlet Allocation (LDA), by complementing it with the relational information (i.e. the network) provided by bigrams (i.e. pairs of two consecutive words) associated to each topic. In other words, we attempt to exploit both the well-established feature extraction ability of topic modeling algorithms and the relation encoding efficacy of network approaches. The idea of associating topic models and network analysis is not new. [3] have suggested encoding the relationship between words and documents in hypergraphs and extracting topics as communities of the hypergraph. Yet, our approach is different, as we build a network for each topic in the corpus starting from the LDA outputs and the bigram distribution containing information about the word order in the text.

To better comprehend the ultimate goal of this project, it is essential to stress that the leading idea is not to study the topic modeling *per se* but to introduce a novel way to enhance topic model results and facilitate the interpretation. That in turn means the goal has a methodological character. Actually, the choice of the Cord-19 corpus was made strategically to serve such a purpose. The richness, variety, and complexity of medical literature related to COVID-19, made it an ideal (and very demanding) setting to apply the proposed method and see if the results were meaningful from an expert's perspective. However, it can be exploited for healthcare literature surveying tasks [4] or bibliometric review [5] ones.

The contribution of this paper is twofold; we provide (i) an intuitive method enhancing the interpretation of topics based on networks of words exploiting bigram information and (ii) a set of heuristics to extract and label subtopics automatically from highly modular topics. The ultimate aim of our approach is to increase the interpretability of topics and, in turn, allow an internal semantic structure analysis at a finer level. LDA2Net intends to be an effective and convenient tool to summarise and make sense of topics in a more rapid and human-friendly way, rather rather than relying on a mere list of words. This novel method permits a transparent and intuitive interpretation of single topics through a visual inspection of (topic-specific) word networks. In addition to this, LDA2Net helps work on multiple levels of granularity, enabling the exploration of the subtopics of interest. Indeed, "indications" provided by automatic topic labels facilitate browsing the numerous topics and finding those of relevance for specific interests. Finally, our approach also allowed us to 1) evaluate the specific relevance of words and their associations per topic, by means of network metrics and 2) cluster topics using aggregated statistical measures, to differentiate cross-cutting topics from specialized ones.

The remainder of this paper is organised as follows. In the Related Work section, we give a brief literature review useful to understand the context within which our work is framed. In the section Materials and Methods, we describe our approach while in the Results section we present the research results and findings. Finally, in the Discussion and Conclusions section, we summarize and interpret the outcomes of the work and provide future line research directions. We would like to underline that, for conciseness, we opted to report all the concepts not

strictly necessary to understand the method in the Appendix sections while being careful not to sacrifice effectiveness for brevity.

## Related work

Latent Dirichlet Allocation (LDA) [6] is indisputably the most frequently used topic modeling approach, and, despite its simplicity, it has established itself as the state-of-the-art Probabilistic Graphical Model in numerous applied research fields. Most criticism towards LDA has addressed some of its statistical limitations, such as the lack of unambiguous criteria for choosing the number of topics, the inability to capture correlations between topics, or its static nature. Many solutions to these shortcomings have been proposed in further developments of the basic probabilistic approach to topic modeling. For instance, research has focused on extending or modifying LDA to account for syntax [7], correlations between topics [8], semantic data [9], and metadata [10], mainly to overcome problems related to the simplifying assumptions of the LDA model. However, less attention has been devoted to addressing the limitations of LDA and related models, concerning their "bag-of-words" approach, which neglect word order. These models return a (weighted) list of words for each topic, disregarding the short-distanced semantic information in the word arrangement sequences.

Even when extended to consider *n-grams*, that is sequences of *n* consecutive words, rather than words as basic units of analysis, it returns a weighted list of such entities, without considering their interrelations. Attempts to include and model syntagmatic information (i.e. information concerning sequential relations between words) in topic models have already been investigated in [11–15] as well. Moreover, using a Dirichlet distribution implicitly assumes that few items overwhelmingly contribute to each topic. This leads to using a topic's top-ranking words to summarise its content. However, the interpretation of such short lists of words is limited, as the weights offer no semantic information regarding the structure of the topic and how words are related to each other.

The proposed topic model enrichment method, named `LDA2Net`, allows to incorporate in LDA bigram information without any statistical assumption on the data generation process for word sequences, being based on observed document-level bigram counts. Differently from other works [16–19] that aim to improve the coherence and interpretability of inferred topics by using domain knowledge, including Linked Open Data, Knowledge Graph Embeddings, and ontologies from the the semantic web, the topic model enrichment method proposed in this work does not rely on any information or data external to the corpus.

Even though the method is here applied only to the LDA model, it can be easily implemented with other topic models, such as correlated topic models CTM [8], or structural topic models STM [10]. That means our approach can be cast into any topic model framework based on the bag-of-words paradigm to transform the topic's distributions of unigrams into topic graphs. It is worth noting the project's primary focus is not on topic modelling *per se* but enhancing the topic modelling. That is possible through our novel deterministic method, based on bigrams, which takes place after the topic inference stage (i.e., unigrams-by-topic distributions). In other words, bigrams are used to enrich an already estimated topic model and to summarize the syntagmatic structure of inferred topics. For the sake of demonstration, in this work, we will be utilizing only LDA as a basic illustrative example, precisely chosen for its simplicity and widespread use.

Concerning topic models estimated with bigrams [11–15], one of the advantages of `LDA2Net` is that its results do not depend on any assumption on the distribution family and data generation process of bigrams, being `LDA2Net` based on the observed frequencies of

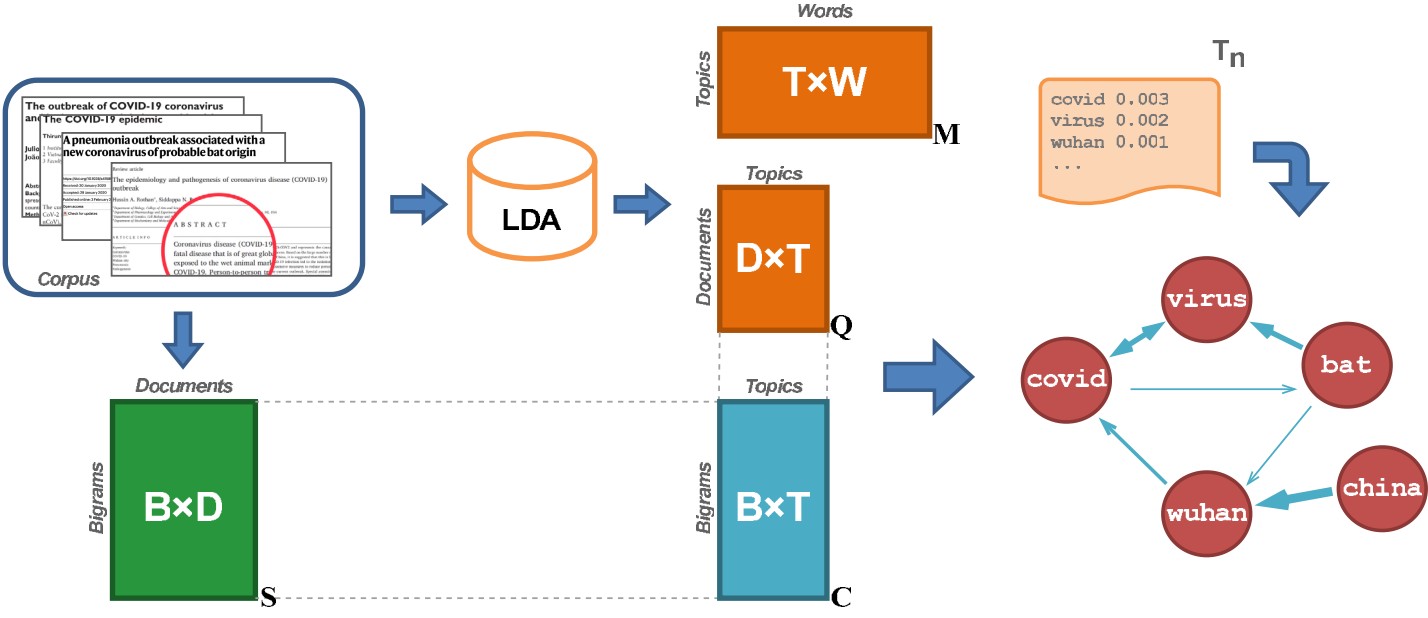

**Fig 1. From LDA output to networks, a summary diagram.**

bigrams in documents, which are combined (a-posteriori) with LDA output matrices, as shown in Fig 1.

## Materials and methods

### Dataset

The current work is based on article abstracts of the Cord-19 dataset (Version 93: 21-06-2021), which is made available by Semantic Scholar and the Allen Institute for AI. For details on the Cord-19 corpus please see [1].

The dataset contains coronavirus-related research papers. Papers in the Cord-19 corpus were sourced from PubMed, bioRxiv, and medRxiv, plus articles from a repository of more than two hundred thousand papers maintained by the World Health Organization. As this work intends to focus on COVID-19, abstracts uploaded before December 31, 2019 (date in which the WHO was first-informed of COVID-related cases of pneumonia in Wuhan) have been filtered out. The resulting dataset contains 398818 abstracts. This conservative filtering approach was adopted for ensuring that the articles selected were post-dating the outbreak's formal acknowledgment and to maintain the representativeness of the CORD-19 corpus in our work.

Details about the preprocessing stage can be found in Appendix B while information about the dataset in Table 1.

**Table 1. Dataset summary information and LDA parameters.**

|  | # documents | # topics | # words (vocabulary) | # bigrams (vocabulary) |
|---|---|---|---|---|
| *value* | 398818 | 120 | 34563 | 4075820 |
| *notation* | $N_D$ | K | $N_W$ | $N_B$ |

## From LDA to topic networks

**Latent Dirichlet Allocation.**   This section outlines the basic concepts about Latent Dirichlet Allocation (LDA) needed for understanding our approach. As LDA analysis is not the crux of this work, we refer the reader to [6] and Appendix A.1 for a thorough introduction to LDA, and to Appendix A.2 for further details about the model estimation and parameters selection. Additional supplementary materials are available online at the following link: https://github.com/carlosantagiustina/underthesurfaceofCOVID19topics.

In layman's terms, LDA is an algorithm that reads through some text documents and automatically outputs the topics therein contained. In order to perform this process, LDA takes as input a collection of documents $D$—where $|D| = N_D$—called *corpus*. Each document being represented as a set of words belonging to a vocabulary $W$, namely the list of all unique words in the *corpus*. Formally, $W = \{w_1, \ldots, w_{N_W}\}$, where $N_W = |W|$. In this context, a topic is a set of words that occur frequently together. The number of topics $K$ to be found is instead an input parameter. With a view to processing the documents, the *corpus* is transformed into a matrix containing the counts of words by document, hereafter called $\mathbf{U}$, of size $N_D \times N_W$. Each entry $\mathbf{U}_{d,i}$ represents the number of times the word $w_i$ appears in the document $d$, where $i = 1, \ldots, N_W$ and $d = 1, \ldots N_D$.

LDA outputs are distributions namely:

- a distribution of words for each topic (i.e. a weighted list of words);

- a distribution of topics for each document.

In [20] a topic is formally defined as a distribution over a fixed vocabulary of words. These results are usually delivered in matrix format. Notably, let $\mathbf{M}$ be the topic-word distribution matrix of size $K \times N_W$ and $\mathbf{Q}$ be the document-topic distribution matrix of size $N_D \times K$. Then, the entry $\mathbf{M}_{k,i}$, where $k = 1, \ldots K$ and $i = 1, \ldots N_W$, is the weight of the word $w_i$ within the topic $k$, whereas the entry $\mathbf{Q}_{d,k}$, where $d = 1, \ldots N_D$, is the proportion of topic $k$ in the document $d$. For a visual interpretation of these matrices, we refer the reader to Fig 1.

**Network construction.**   This section explains how to convert any topic obtained through LDA into a network. Basically, we aim to convert a weighted list of words into a weighted and directed network, a network where each edge has both a direction and weight, and every node has a weight. For further definitions and notation about network theory, we refer the reader to Appendix C.

The ultimate goal is thus finding a way to define the adjacency matrix of the network of a chosen topic. In other words, we need a way to gauge the direction and the weight of the syntagmatic relation between any pair of words represented by edges in the network. As first stage, the `LDA2Net` method requires a data preparatory phase that involves collecting all bigrams present in the *corpus* and arranging them in a fashion akin to the matrix $\mathbf{U}$. That means, building a vocabulary for bigrams $B = \{b_1, \ldots, b_{N_B}\}$, where $N_B = |B|$. Unlike $W$, this vocabulary contains unique ordered pairs of words. Then, bigram frequencies by document are collected into a matrix $\mathbf{S}$ of size $N_B \times N_D$. Thus, each entry $\mathbf{S}_{b,d}$ represents the frequency of the bigram $b$ in the document $d$. The next stage involves computing one of the two components of the weights associated with a bigram, conditioned by the topic under analysis. This component is called *counts − weight*. Given a topic, a bigram will assume a *counts − weight* proportional to both its occurrence (counts) in documents and the proportion of the given topic in documents. This, in turn, means the same bigram may have a different *counts − weight* depending on the topic under investigation. Formally, let $\mathbf{C}$ be the matrix of size $N_B \times K$ collecting bigram *counts − weight*s by topic; then each entry of $\mathbf{C}$ is computed as $\mathbf{C}_{b,k} = \sum_d \mathbf{S}_{b,d} \mathbf{Q}_{d,k}$. The same definition

expressed for the whole matrix **C** is:

$$\mathbf{C} = \mathbf{SQ} \tag{1}$$

However, the quantities in **C** do not take into account the topic-specific probabilities of words composing the bigram, inferred through LDA. In fact, this information is given by **M**, the topic-word distribution matrix. For this reason, in the computation of the network's adjacency matrix, we combine the *counts − weight* with a second component, called *probs − weight*, which embeds these very values. Specifically, let $A^k$ be the adjacency matrix describing the word network for the topic *k*, of size $N_W \times N_W$. Regardless of the topic under consideration, you may notice that every network has the same node-set. Nevertheless, topic networks are not identical as the weights of edges between words and the node weights distinctively characterize each topic.

In this regard, let $e_{i,j}^k$ be the weight of the bigram made by the ordered pair of words $w_i$ and $w_j$ in the network representing topic *k*. Such a quantity exists if and only if the pair of words $(w_i, w_j)$ is a bigram belonging to the vocabulary of (observed) bigrams *B*.

Then, we define the adjacency matrix for the topic *k* as:

$$A_{i,j}^k = \begin{cases} 0, & \text{if } (w_i, w_j) \notin \mathcal{B} \\ e_{i,j}^k, & \text{otherwise} \end{cases} \tag{2}$$

where $e_{i,j}^k$ is given by the product between the *counts − weight* of the bigram $(w_i, w_j)$ for the topic *k*, that is $\mathbf{C}_{b,k}$, and its *probs − weight*, obtained by multiplying the LDA word probabilities of $w_i$ and $w_j$, which are respectively given by $\mathbf{M}_{k,i}$ and $\mathbf{M}_{k,j}$. As a result, we have that `LDA2Net`-weight of the bigram $b = (w_i, w_j)$ is:

$$e_{i,j}^k = \mathbf{C}_{b,k}\mathbf{M}_{k,i}\mathbf{M}_{k,j} \tag{3}$$

In this respect, we would like to point out that if the bigram composed by $(w_i, w_j)$ does not exist, it does not imply the bigram $(w_j, w_i)$, in turn, does not exist, i.e. $(w_i, w_j) \neq (w_j, w_i)$. In fact, a bigram represents an oriented edge between two words. Finally, we normalize $A^k$ entries so that, for each *k*, the sum of all entries is equal to 1.

## Automatic topic labelling

In this section, we present a heuristic for generating labels for topics and sub-topics by leveraging the network structure and the edge and node metrics described in Appendix C. Broadly speaking, node/edge metrics or centralities are indices over nodes calculated by accounting for the topological characteristics of the network. The resulting labels are obtained through random walks over the networks and are in the format of a sequence of words. A visual outline of the this heuristic is shown in Fig 2 p. 6.

Let *m* be a community, which is a subset of nodes within a graph such that connections between nodes are denser than connections with the rest of the network. Let $\mathbf{G}_m^k$ be the subgraph of the community *m* within the network representing topic *k*. A node is chosen to be the starting vertex of a random-walk based on the out-degree centralities. This node is sampled from a discrete distribution for which the probability of node *i* being extracted is proportional to its weighted out-degree, that is $deg_i^{out}$. In other words, given a community network $\mathbf{G}_m^k = \mathbf{H}$,

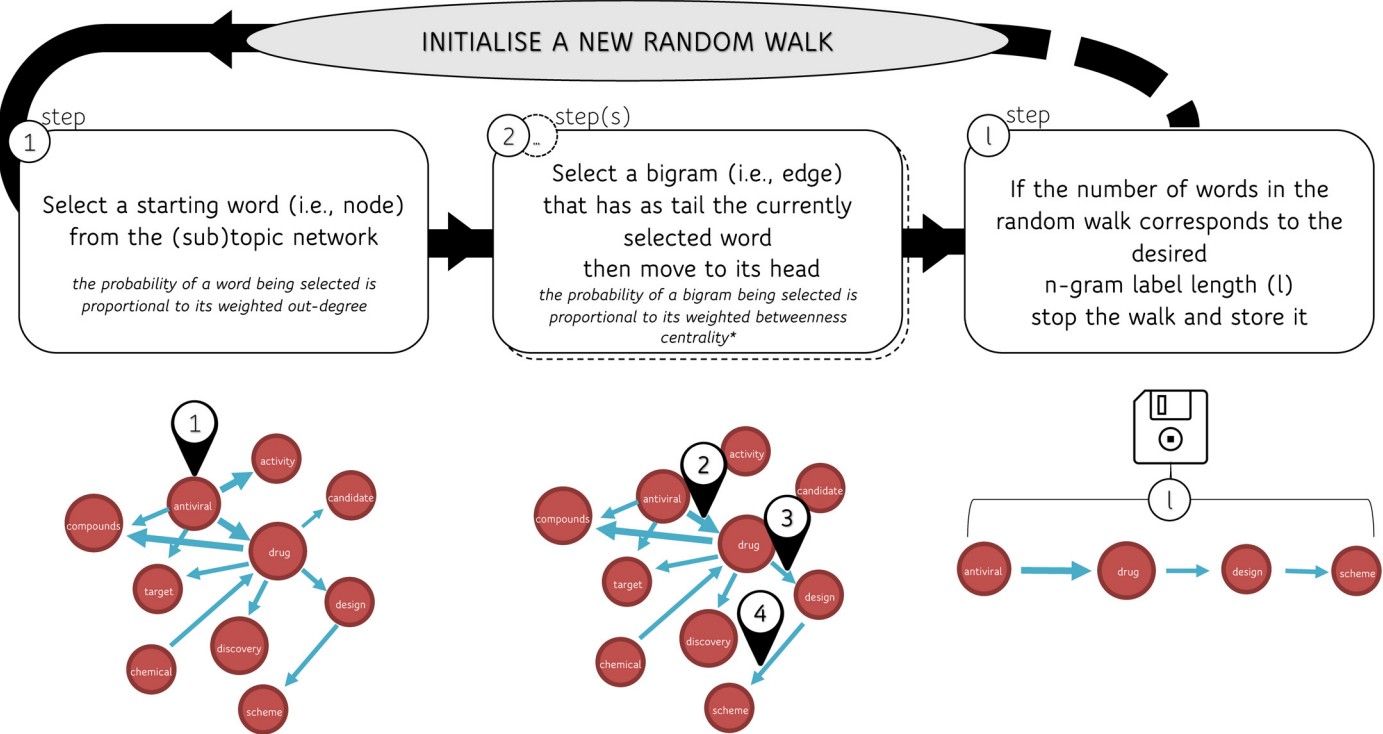

**Fig 2. Visual summary of the heuristic employed for generating (sub)topic label proposals.**

the probability $Pr_{i_{\mathbf{H}}}$ of a node $i$ to be selected is:

$$Pr_{i_{\mathbf{H}}} = \frac{deg_i^{out}}{\sum_z^{|\mathbf{H}|} deg_z^{out}} \qquad (4)$$

One of the advantages of this initialization step is that the random walks are more likely to start from nodes that are hubs (i.e. nodes characterized by a high degree centrality) from the point of view of outgoing edges, which is a good proxy for being a starting term in phrases related to a given topic or subtopic (where a subtopic is actually a community).

Once the starting node has been sampled, the random walker begins to wander through the network, basing its direction choices on a specific criterion: the edge betweenness centrality. The edge betweenness centrality is the edge counterpart of the node betweenness centrality and identifies edges in the network that are crucial for information flows. The betweenness centrality of an edge between nodes $i$ and $j$, $btw_{(i,j)}$, actually captures the overall relevance of a bigram in the whole network, thus providing a global rather than local criterion for randomly moving from word to word. Given the selected starting node (word) $i$, the probability of moving from $i$ to $j$ is computed as follows:

$$Pr_{(i,j)_{\mathbf{H}}} = \frac{btw_{(i,j)}}{\sum_z^{|\mathbf{H}|} btw_{(i,z)}} \qquad (5)$$

In case none of the edges having $i$ as tail have a positive betweenness centrality, edge weights $e_{i,j}$, are used instead of the betweenness centrality $btw_{(i,j)}$ as criterion for randomly

moving from node to node. The random walk is halted if the selected word $i$ has no outgoing edges.

Given a desired length $l$ of the $n$-gram label, through our procedure, one can generate thousands of random walks (i.e., $n$-grams) of the same targeted length, then order them by relative frequency and use the most frequently sampled one(s) as label candidate(s) for a topic (whole network) or a subtopic (i.e. community, a subgraph of the network). The proposed heuristic does not incorporate information external to the corpus and is not based on semantics. Instead, it leverages graph properties, like the interconnectedness of words within the subtopic networks, to generate label candidates of varying lengths. It is important to note that our method focuses on statistical associations and structural relationships, not meaning inference. While the method does not engage in reasoning, it does provide an automated means of suggesting labels based on the analyzed structural properties of word networks.

## Results

In the following sections, we present a very-compact summary of relevant aspects of the proposed methodology together with associated results and key findings. For an exhaustive visual recap of each topic, we refer the reader to supplementary material available at this Link.

The following results are based on a LDA model with 120 topics estimated using a subset of the abstracts of the Cord-19 corpus. It is worth stressing that, as LDA results depend on the algorithm's random initialization, the proposed method also inherits such a randomness aspect. That means different initializations will provide different topics and, in turn by applying our approach different graphs. Yet, given the algorithm's robustness, the results will tend to be very similar. For further details about LDA implementation please see A.1. In order to have a sort of ground-truth/baseline for our method, we asked an expert to assign a label to each topic, only by considering its top 25 words (by LDA probabilities). Labels are reported in Table 2. Specifically, given the peculiarity of this corpus and the strong presence of technical terminology, we chose as person performing such a labelling a figure with a recognised biological and medical background.

### Topics as networks

The first contribution of this work concerns the readability of topics. LDA can be used to discover topics in an extensive collection of documents and provides a (weighted) list of words for each topic. However, the interpretation of these word lists is often tricky and arbitrary as LDA does not provide any information on word associations and sequences. Our approach addresses this problem by transforming topics, i.e. weighted list of words obtained via LDA, into (weighted and directed) networks capable of capturing (short-distance) word relations. Basically, LDA2Net converts each LDA topic into a network where nodes represent words and edges represent relations between them. Such a conversion is carried out by applying

**Table 2. Most frequently sampled label candidate for the four subtopics of topic #88.** In parenthesis frequency of the walk out of a sample of 1000 random walks of that length.

| subtopic# | 2-gram label | 3-gram label | 4-gram label |
|---|---|---|---|
| 1 | drug→target (118) | antiviral→activity→compounds (42) | antiviral→targets→identified→compounds (29) |
| 2 | main→protease (220) | main→protease→pro (66) | binding→interactions→evidenced→AAs (34) |
| 3 | viral→RNA (346) | viral→RNA→plus (21) | viral→RNA→plus→nucleocapsid (14) |

AAs is the acronym for Amino Acids and RNA is the abbreviation of Ribonucleic Acid.

weights to edges between words, representing the strength and direction of their sequential arrangement in the corpus. These weights are based on the combination, through matrix multiplication, of observed frequencies of bigrams in documents and LDA's output matrices (see Fig 1). In essence, `LDA2Net` makes topics more transparent and readable by binding their interpretation through observed word associations.

For the sake of brevity and illustrative purposes, for this section, we chose only a few topics to show our results, in particular, topic #50 and topic #88 in Figs 3 and 4, respectively. Based on the expert's evaluation (that is, according to the label assignment by an expert to each topic of our model by considering the LDA list of top 25 words only), topic 88 is about antiviral drug molecules. In contrast, topic 50 has no explicit subject, and thus, the label is not defined. In these two figures, we compare the list of the top 25 words obtained through LDA (Tables 3 and 4) sorted by topic-word probabilities and the graph of the top 25 bigrams by `LDA2Net` weights. As regards graphs, edge widths are a function of topic-specific bigram weights obtained through `LDA2Net`. In other words, the edge width is proportional to the edge weight, the tie strength between those two words in that topic. The two figures highlight the interpretive advantage offered by the proposed method. For instance, by observing the network of topic #50 (Fig 3) we can immediately notice the marginal role played by the word `CI`, which by contrast is the most important one by the topic-word distribution obtained through LDA. Interestingly, the word `risk`, which is ninth in terms of probability, appears to play a relevant role in the bigram network. A reader using only probabilities would probably focus her attention on the confidence interval acronym (`CI`), possibly missing that the core issue of the topic is the measurement of COVID-related risk factors. The network built over the bigrams of topic #88 (Fig 4) contains many disconnected components, suggesting that this topic is likely modular, that is to say there are many subtopics, each representing separable but related aspects of topic #88. For example, while `binding→affinity` and `viral→replication` are distinct issues they are often mentioned together in the corpus. As both figures prove, the networks do actually ease the interpretation of the topic and allow

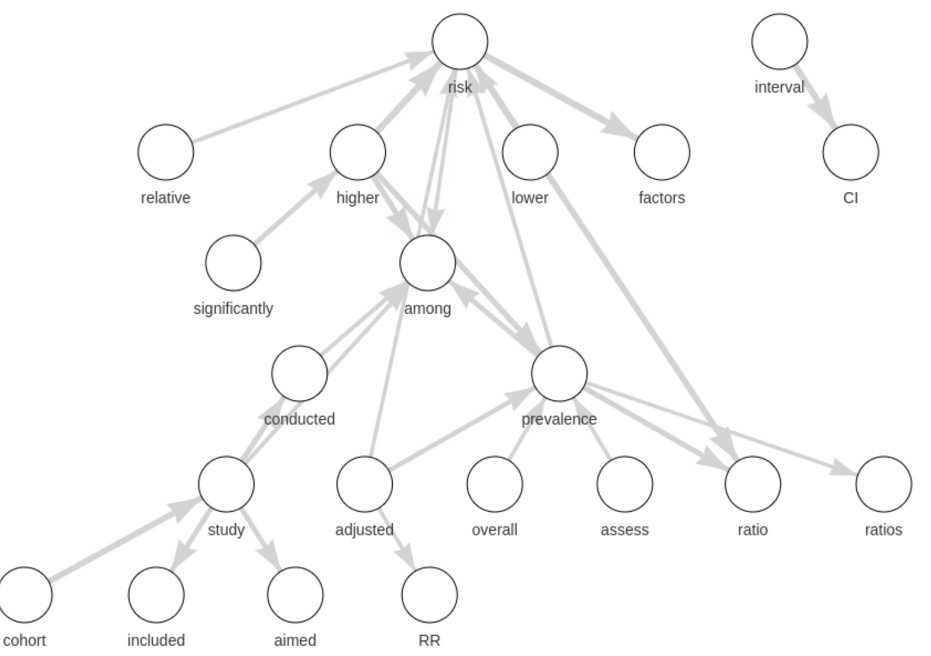

**Fig 3. Topic #50: Network of top 25 bigrams by `LDA2Net`-weight.**

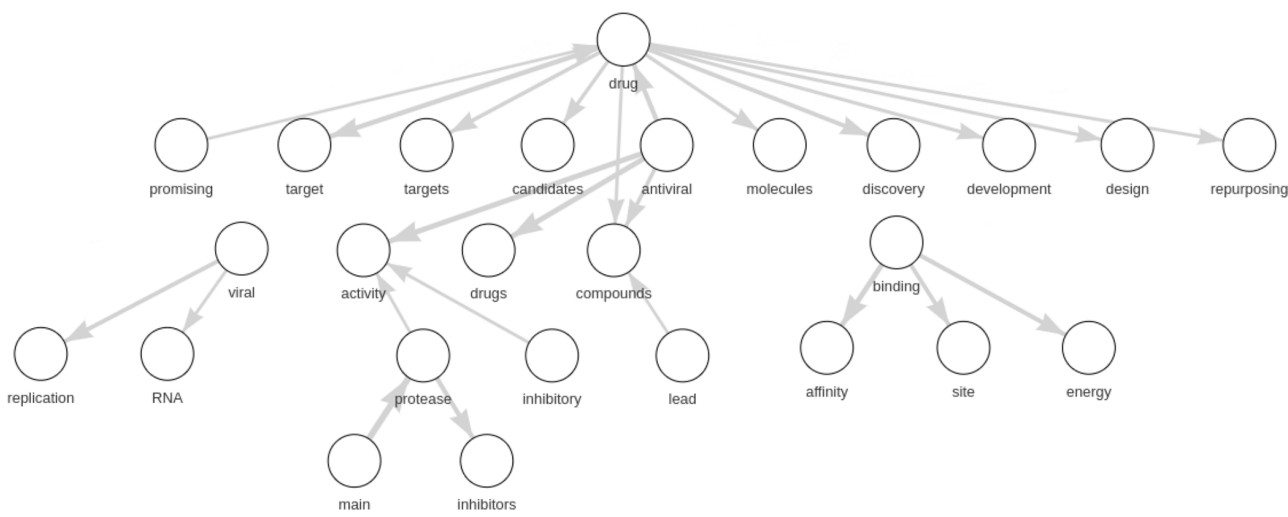

**Fig 4. Topic 88: Network of top 25 bigrams by `LDA2Net`-weight.**

**Table 3. Topic #50: Table of first 25 words ranked by LDA probabilities.**

| rank | unigram | prob. |
|---:|---|---:|
| 1 | CI | 0.2283 |
| 2 | prevalence | 0.0446 |
| 3 | ratio | 0.0380 |
| 4 | among | 0.0378 |
| 5 | interval | 0.0267 |
| 6 | adjusted | 0.0250 |
| 7 | study | 0.0227 |
| 8 | association | 0.0209 |
| 9 | risk | 0.0198 |
| 10 | compared | 0.0180 |
| 11 | RR | 0.0172 |
| 12 | higher | 0.0163 |
| 13 | respectively | 0.0158 |
| 14 | included | 0.0142 |
| 15 | analysis | 0.0113 |
| 16 | aor | 0.0110 |
| 17 | analyses | 0.0109 |
| 18 | multivariable | 0.0105 |
| 19 | ratios | 0.0105 |
| 20 | conducted | 0.0093 |
| 21 | estimated | 0.0087 |
| 22 | age | 0.0086 |
| 23 | total | 0.0083 |
| 24 | overall | 0.0082 |
| 25 | cohort | 0.0077 |

**Table 4. Topic 88: Table of first 25 words ranked by LDA probabilities.**

| rank | unigram | prob. |
|---|---|---|
| 1 | compounds | 0.0289 |
| 2 | binding | 0.0226 |
| 3 | drug | 0.0220 |
| 4 | activity | 0.0173 |
| 5 | drugs | 0.0163 |
| 6 | inhibitors | 0.0162 |
| 7 | antiviral | 0.0152 |
| 8 | protease | 0.0151 |
| 9 | target | 0.0145 |
| 10 | main | 0.0111 |
| 11 | molecules | 0.0108 |
| 12 | viral | 0.0101 |
| 13 | targets | 0.0090 |
| 14 | replication | 0.0089 |
| 15 | mpro | 0.0083 |
| 16 | site | 0.0076 |
| 17 | inhibitor | 0.0076 |
| 18 | RNA | 0.0070 |
| 19 | inhibition | 0.0066 |
| 20 | affinity | 0.0065 |
| 21 | promising | 0.0064 |
| 22 | compound | 0.0063 |
| 23 | potent | 0.0063 |
| 24 | novel | 0.0062 |
| 25 | interactions | 0.0060 |

to unveil the word organization behind each topic—which would not be identified through the use of mere word lists.

## Word centralities

Here we explore the importance of words in each topic based on metrics derived from network analysis. These peculiar measures are called centralities as they are indicators of a node's centrality (importance) within a graph. In practice, by accounting for the topological characteristics of the network and each node position within it, centrality measures permit to rank vertices by structural relevance within a network. In this work, we consider in particular the following centralities: degree, in-degree, out-degree, page-rank and betweenness centrality. Degree centrality counts the number of neighbors a node has. We have two further versions of the measure if the network is directed: in-degree and out-degree. Intuitively, in-degree is the number of links pointing inward at a node while out-degree is the number of connections originating at a node and pointing outward to other vertices. The PageRank centrality defines the importance of a node by considering the number of links it receives, the link propensity of the linkers, and the relevance of the linkers. Finally, the betweenness centrality, one of the most popular measures of the influence of a node, scores vertices based on their connectivity by counting the paths connecting nodes to each other. This metric assumes that important vertices are bridges over which information flows; then, if information spreads via shortest paths,

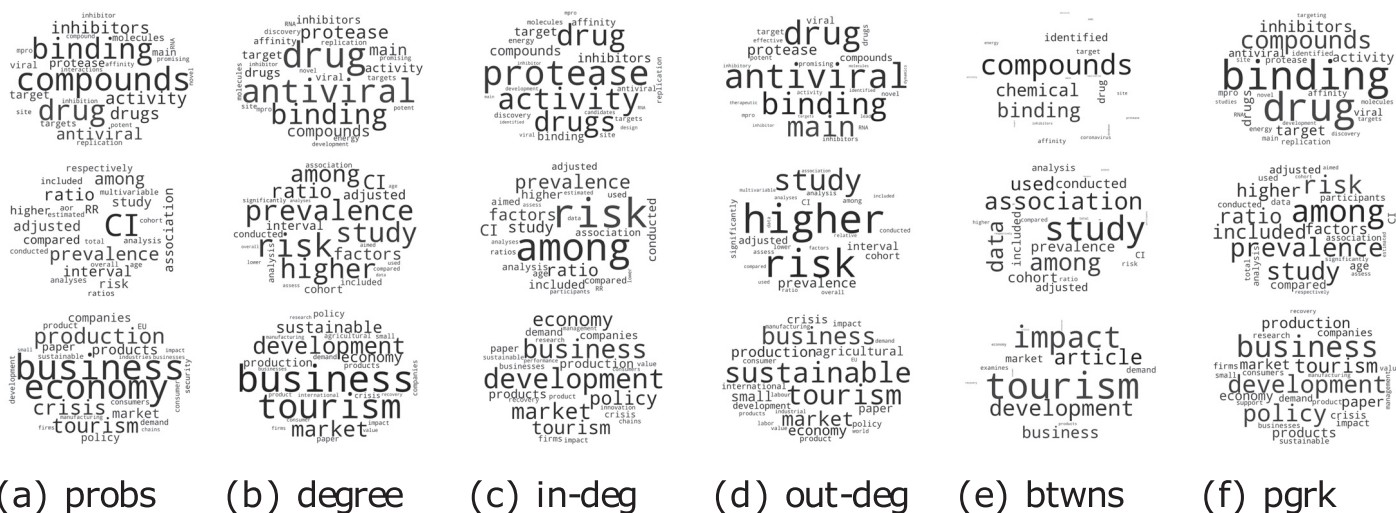

**Fig 5. Top 25 words (i.e., nodes) by centrality measure, for topics #88 (upper row), #50 (middle row) and #36 (lower row).**

important nodes are on those paths. That means by counting the number of these paths, we get an estimation of the importance of a node. Further details about centralities can be found in C.

Recall that via `LDA2Net` we generate a network where nodes represent words, and edges represent directed relations between pairs of words (i.e. two nodes connected by a directed link express a bigram). Like in network analysis, we can measure the importance of a node employing specific measures; likewise, we can assess the influence of a word within the network of a topic via the same approach. Basically, for each word in a network, where the network typifies a topic, we derived a score specific to that given word and topic. That means we might observe different values for the same word, depending on the topic network it belongs to. By exploiting these relevance measures, we intend to capture different aspects of words based on the underlying network structure. To help the reader, we provide a simplified visualization of these metrics by depicting words' centralities through word clouds, as shown in Fig 5, p. 10—the bigger the word's font size, the greater the word's relevance. Except for the first column, which refers to the plain LDA probabilities, each column shows a centrality measure, while rows denote the topics. In particular, in Fig 5, we examine topics # 88, 50 and 36. According to the expert's evaluation, the last one, topic # 36, concerns the subject of economic impact. In Fig 5 a clear trend stands out: different metrics sharply seize different relevance aspects. For instance, by looking at topic #50, in the middle row, we notice that all network-based metrics de-emphasize the role of the term `CI`, by contrast providing complementary perspectives centered on associations to the word `risk`. One could say that the betweenness centrality helps make sense of the topic's relational context, the in-, and out-degree centralities allow to single-out directional relations between words. The PageRank combines, by construction, both the local information provided by degree measures and the global influence of a word within the network. Indeed, similarly, we can see, for topic #36 (bottom row), that the betweenness centrality puts the focus on one of the main issues contextual to the economic impact (that is, the impact on the tourism sector) while PageRank points out more than one aspects. These observations about centralities make us believe that the structural representation of a topic has the potential to catch many shades of the subject matter. Thus it provides a more meaningful view of the topic.

## Relations between measures

To assess the enrichment effect of the proposed method, we performed a correlation analysis between word centralities. Here, the goal is to figure out which relations hold between the different measures and understand what happens to information when a list of words is being transformed into a network (e.g. redundancy or loss of information). Specifically, we calculated both the Pearson and the Spearman's rank correlation, as shown in Fig 6.

The Pearson correlations show that the centrality metrics preserve the information contained in LDA probabilities. Indeed, the Pearson correlations between LDA probabilities (*probs*) and the other measures (*degree*, *in-degree*, *out-degree*, *betweenness* and *PageRank*) are quite significant, in particular with respect to PageRank and degree centrality. That means the information obtained by both approaches is not so distant. Instead, the betweenness centrality has the weakest correlation with LDA probabilities. That is likely because it offers a different perspective on topic-specific, most relevant words, privileging words that have important structural roles. The Pearson correlation behaviour slightly changes when we consider a smaller set of words, i.e. only the top 30 words by LDA probabilities. On the contrary, the rank correlation values are pretty low when all words are taken into account, whereas they increase when the set of words is reduced. This is assumably due to the "tail" of the word ranking, namely, the lowest positions in the word ranking are interchangeable because word scores are minimal and close to each other. Indeed, when the rank correlation is computed over a smaller set of words, results are more consistent and aligned with Pearson correlation ones. Again, we observe that the betweenness centrality has the weakest correlation, confirming our assumption that it provides a different interpretation of word importance.

## Bigram metrics

One can gain additional insights into the corpus through `LDA2Net` by performing an analysis not only at word level but also at the word association level, associations here captured by bigrams. To this end, in Fig 7, we compare different bigram weighting strategies, that is the *counts − weight*, the *probs − weight* and their combination, actually the weights adopted in our approach, called `LDA2Net − `*weight*, as described in till exploiting the word cloud representation, in Fig 7, we show bigrams in the format `[word1]>[word2]`, for instance `respiratory>syndrome`. Again, the bigger the font size, the more significant the bigram; each row refers to a topic, and each column identifies a given weighting strategy.

As the figure shows, both *probs − weight* and *counts − weight* seem to exhibit some weaknesses. On the one hand, *probs − weight*, which is based on the pairwise product of LDA word probabilities, suffers from over-representing the combination of a few words with high probabilities. On the other hand, *counts − weight*, which is based on document-bigram counts assigned to a given topic, appears to over-represent associations between generic words that are in most documents of the corpus, such as `COVID-19>pandemic` and `coronavirus>pandemic`. Actually, the combination of both strategies into the proposed `LDA2Net − `*weight* summarises well the distinctive word associations characterising each topic. For instance, for the topic #88 (top row) we can note that `LDA2Net − `*weight* highlights not only the bigram `main>protease` but also the bigram `antiviral>drugs`, which is actually the label assigned to this topic by our medical expert. This investigation about edge weighting strategies confirms that the use of networks to reproduce word associations is indeed a practical approach akin to the human interpretation process.

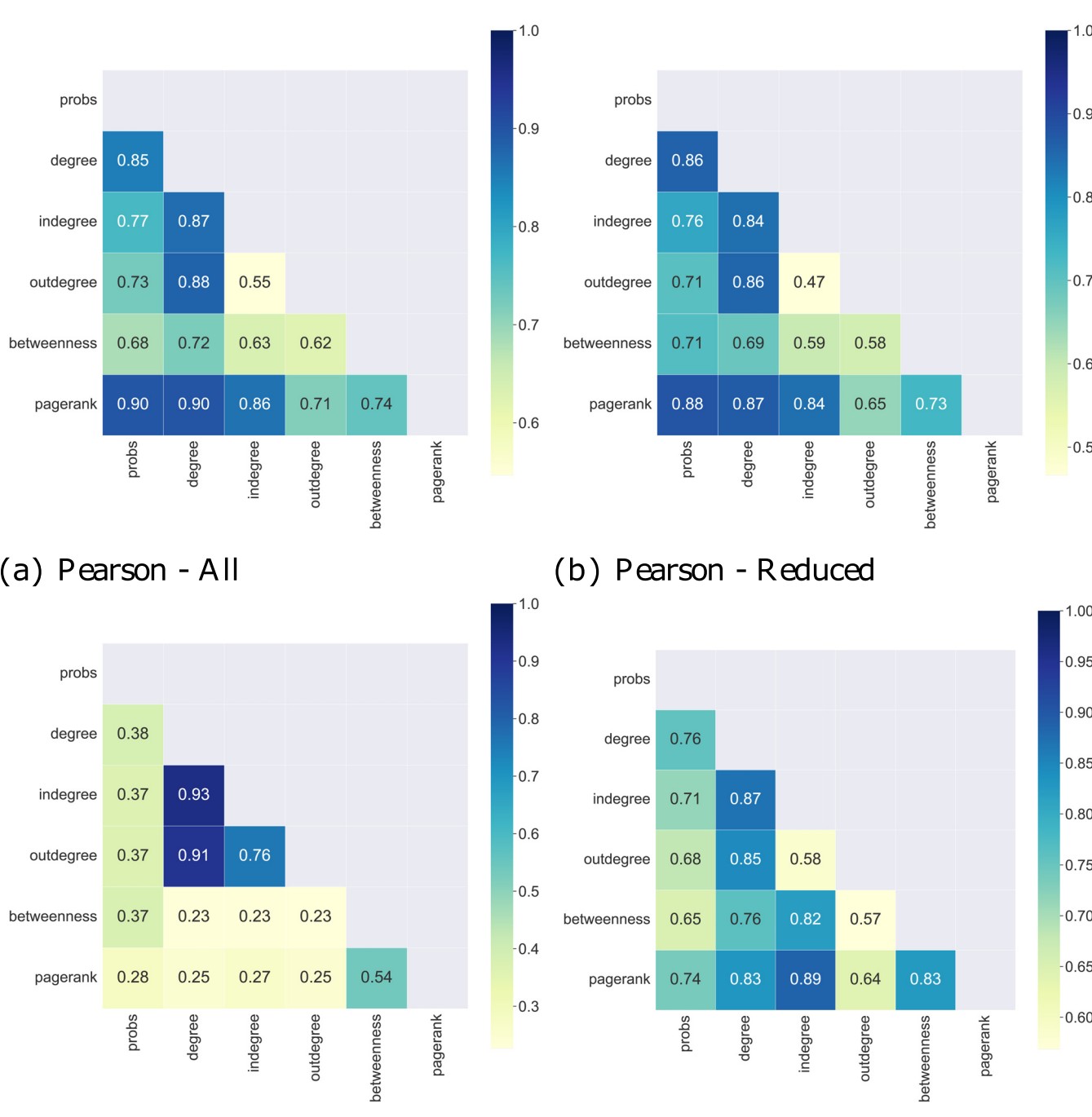

**Fig 6.** Correlation analysis: a) and b) Pearson—c) and d) Spearman's rank correlation between word (node) centralities measures computed over networks and word probabilities from LDA output. a) and c) report correlations computed over all words while b) and d) report correlations considering only the top 30 words by LDA probability.

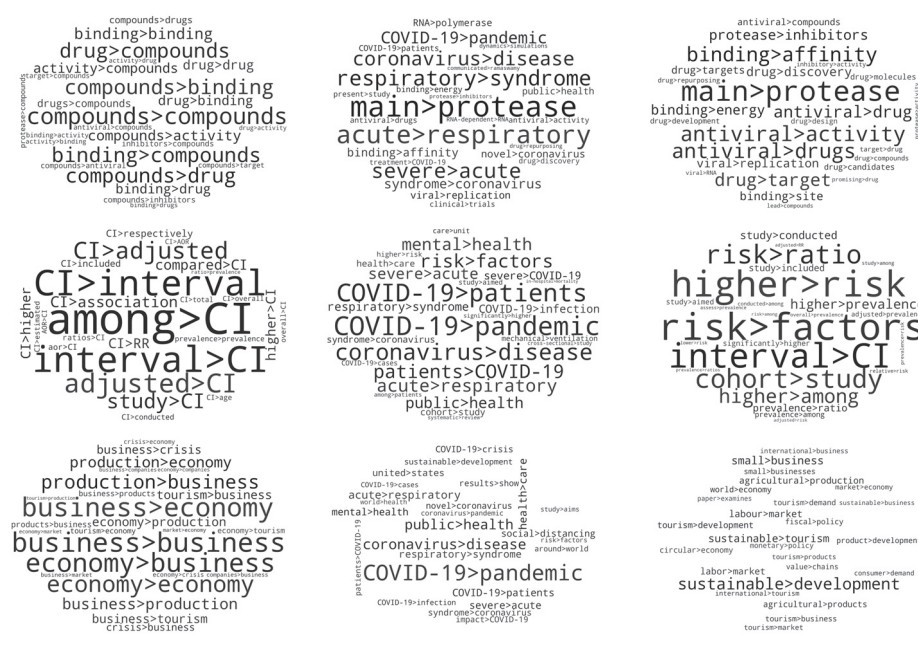

**(a) probs-weights    (b) counts-weights    (c) LDA2Net-weights**

**Fig 7. Top 25 bigrams (i.e., edges) by measure, for topics #88 (first row), #50 (second row) and #36 (third row).** For details about edge weights see Section From LDA to Topic Networks.

## Information gauging

Another way to measure the information added by the `LDA2Net` method is reasoning in terms of divergence that measures the similarity between two distributions. We can determine this measure by calculating the Jensen–Shannon Divergence (JSD) between the two components employed to compute the *LDA2Net − weights*: i) the *counts − weights*, obtained using bigram frequencies, and ii) the *probs − weights*, calculated as a pairwise product of LDA's word probabilities. The Jensen–Shannon Divergence is the symmetric and smoothed version of the relative entropy. The relative entropy, also referred to as the Kullback-Leibler divergence (KL divergence), measures how distant two distributions are from each other. However, the KL divergence is not a distance metric as it is asymmetric and does not meet the triangle inequality. Conversely, the JSD is a distance metric since it extends the KL divergence by providing a symmetrical score. For a more formal definition of JSD we refer the reader to Section Appendix D. Usually, one of the two distributions, named *P*, represents actual observations; instead, the other, called *Q*, is a model of *P*. In our case, *probs − weights* is the model, *Q*, and *counts − weights* the data, *P*. Indeed, *counts − weights* are based on the actual count of bigrams while *probs − weights* are based on the LDA model probabilities, so they are an approximation of data. If the two distributions diverge, it means that their combination does not result in redundant information, that is, the divergence captures the amount of extra information introduced by `LDA2Net` weights. The information brought by the proposed method is highlighted by the distribution of JSDs by topic, shown in Fig 8. In this histogram, we can clearly see that the distribution is heavily skewed towards the right of the JSD values range, that is [0, 1] (where 1 means completely divergent). It suggests our combined measure *LDA2Net − weights* does infer supplemental knowledge.

## Jensen–Shannon divergence between weight componets

**Fig 8. Histogram of JSDs between the two normalized components used to compute bigram (i.e., edge) weights in topic networks.**

### Mapping topics

Thanks to the `LDA2Net` framework, networks representing topics obtained by LDA can be summarized at a macro level by means of a set of topic-specific measures allowing to distinguish different classes of topics based on some key properties. The goal of this section is to investigate such a grouping behavior.

We employed the following measures to characterize topics:

- *Topic Mean and Variance*: respectively the average value $\mu_k$ and its variance $\sigma_k^2$ of the probability of a topic $k$ to appear in the corpus, formally

$$\mu_k = \sum_d \mathbf{Q}_{d,k}/N_{\mathcal{D}}$$

and

$$\sigma_k^2 = \sum_d \left(\mathbf{Q}_{d,k} - \mu_k\right)^2/N_{\mathcal{D}}$$

- *Jensen–Shannon Divergence*: that is the divergence between the two parts composing the *LDA2Net – weights* (*counts – weights* and *probs – weights*), already described in *Information Gauging* section, and Appendix D.

- *Barrat Clustering Coefficient* (BCC): the weighted version of the clustering coefficient for each (topic-specific) network [21].

The clustering coefficient is a primary descriptive statistic of networks that measures the cohesion between nodes (not to be confused with the node tendency to form densely connected groups, that is communities). An intuitive example of this descriptor is its application

in the friendship network scope. In this case, the clustering coefficient reflects the extent to which a person's friends are also friends of each other. A more exhaustive explanation of the clustering coefficient is given in Appendix C.2.3. The BCC summarises the network-level properties of the topics, and the Topic Mean and Variance characterise topics by their distribution in the corpus, inferred through LDA. Finally, the JSD takes into consideration both aspects. To cluster topics in an unsupervised way, we adopted the Gaussian finite mixture model, a formal statistical framework on which to base the clustering procedure. In a nutshell, the model assumes a finite mixture of probability distributions generates data, and each cluster follows a different multivariate probability density distribution [22]. One of the main advantages of this approach is that the number of groups (mixing components) and other parameters, such as those about covariance, are selected automatically. All details about the model are reported in Appendix E. As shown in Fig 9, three classes emerge, but one of them is composed of a few elements though. We identified those outliers as linguistic topics, that is, topics containing all words in a given idiom (e.g. topic # 54 contains only French words and #106 only Spanish words). The other two classes seem to be mainly characterized by the relation between JSD and mean and the BCC and variance. In particular, we observe a group where, regardless of the mean, the JSD is above a certain threshold, and regardless of the BCC, the variance is below a specific value. Topics belonging to this class are, for instance, # 119 and 49, 33, which can be classified as cross-cutting topics (green triangles). On the other hand, specialized topics (red squares), that is, topics on a particular subject matter, exhibit higher variance and lower JSD, such as topic #99, 12 and 50 about cellular mechanisms, online education during the pandemic, and the inhibitors of SARS-CoV-2 main protease, respectively.

Results suggest that the documents of the Cord-19 corpus are based on two structurally different sets of topics: a set about more technical topics focused on field-specific terms and another set on cross-cutting topics. In the former, the information provided by the networks is relatively more in line with that provided by LDA's topic-word distributions. For the latter, networks provide further information that cannot be captured by basic topic modeling approaches (e.g. "bag-of-words").

These findings, even though germinal, are encouraging. Indeed, such a classification could be convenient if an LDA user wanted to perform an initial investigation of topics in a fast but still accurate way (for instance, by first considering the specialized topics and only then the cross-cutting ones).

## Detecting and labelling sub-topics

An additional advantage provided by LDA2Net consists in detecting and labeling subtopics. While the labeling strategy has already been explained in *Automatic Topic Labelling* section, here we introduce the detecting procedure. This approach aims to exploit the parallelism between communities in networks and subtopics in topics. In network analysis, a community is formally defined as a subset of nodes, densely connected and loosely connected to the nodes in the other communities in the same graph. The procedure to find these groups of nodes within networks is called community detection. A brief introduction to this argument can be found in Appendix C.2.2. A way to measure the strength of division of a network into modules (also called groups, clusters, or communities) is through modularity. Networks with high modularity have a dense set of connections between vertices within modules but only sparse connections between vertices within different modules. Similarly, a network representing a topic with high modularity will have strong relationships between words within subtopics and weak relationships between words in different subtopics. Then, we can also assume that topics having high network modularity are good candidates for containing subtopics (i.e. closely tied

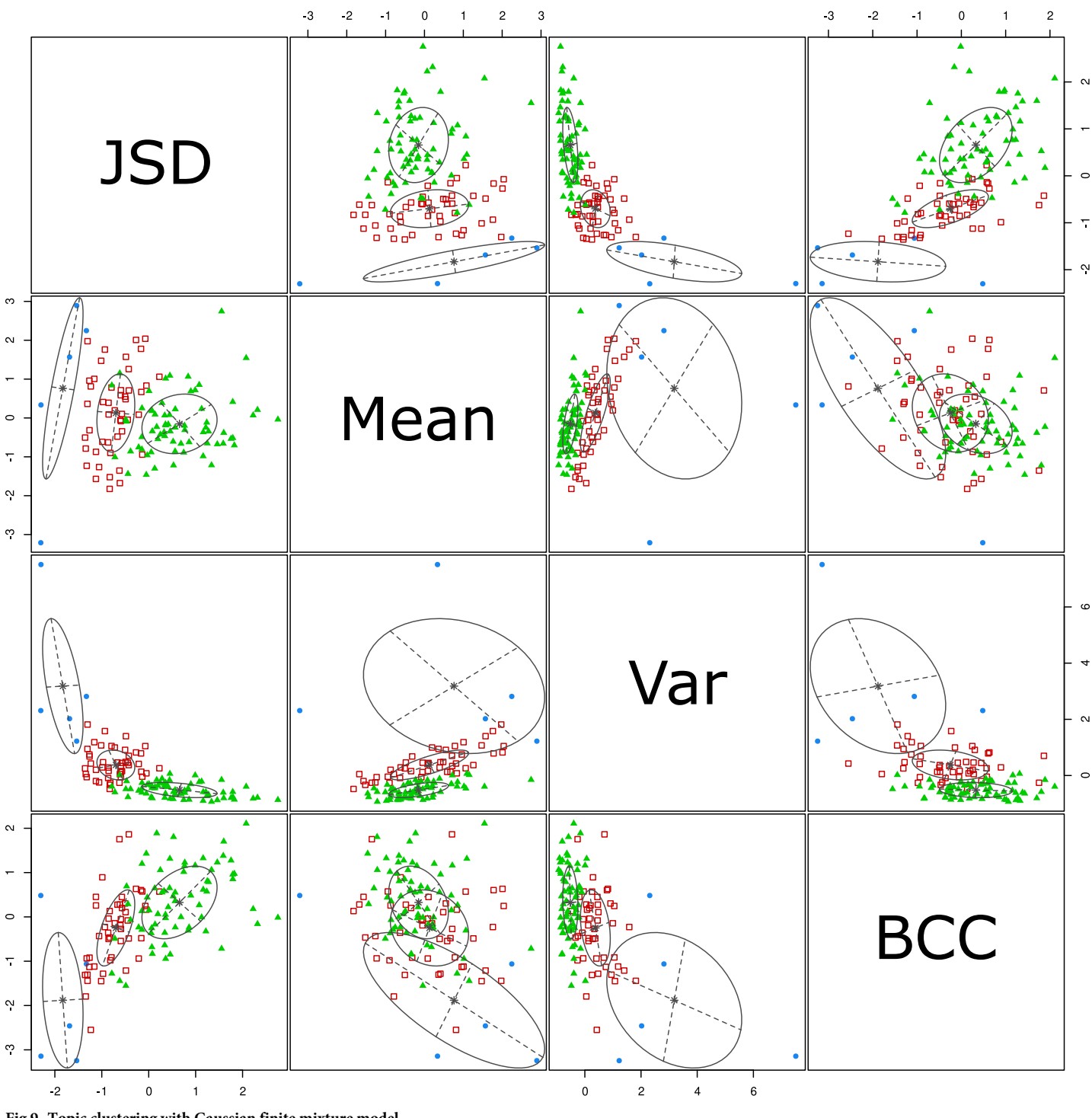

**Fig 9. Topic clustering with Gaussian finite mixture model.**

word communities). In Figs 10 and 11 and we show network modularity values by topic via bar chart and the related distribution, respectively. In particular, in Fig 11 we can notice how the distribution of modularity values is skewed and with a long right tail; about half of the topics have modularity very close to zero, indicating that not all topics contain meaningful

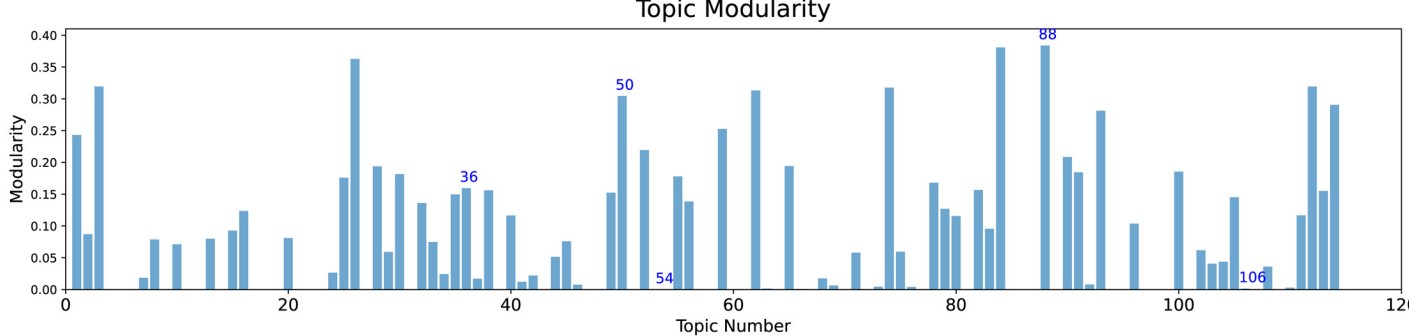

**Fig 10. Bar plot of network modularity values, by topic.**

subtopics. For this reason, in this section, we decided to show results concerning topic #88, selected just because it has the highest modularity and then comprises subtopics. According to our expert, we recall that topic #88 is about antiviral drug molecules. A filtered version of the whole network for this topic is shown in Fig 12. The filtering stage of the networks is necessary for visualization purposes, as the original networks are dense. Specifically, the filtered network has been obtained by sorting words by LDA probabilities and keeping the top 1% percentile words. Also, edges (representing bigrams) were filtered, again by keeping the top 1% percentile based on their *LDA2Net − weights*. We can notice in Fig 12 that the topology of the network clearly reveals a modular structure organized around few hubs, such as the words `binding`, `compounds` and `drug`. To detect communities, we employed the weighted walk-trap method proposed by [23] and described in Appendix C.2.2. We extracted a set of communities through the community detection algorithm, which are essentially sub-graphs and subtopics. An illustrative example of obtained outcomes is given by Fig 13 where three different subtopics

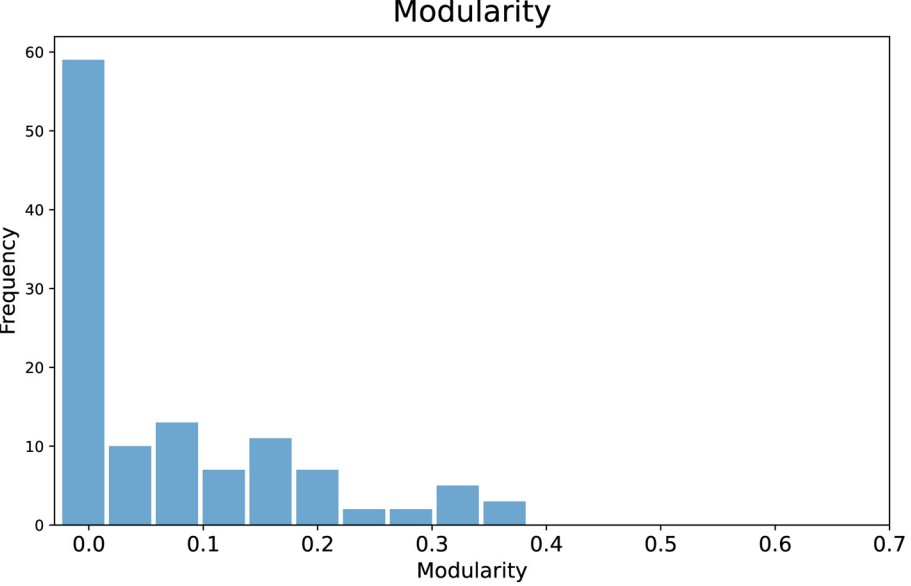

**Fig 11. Histogram of the topic-network modularity values.**

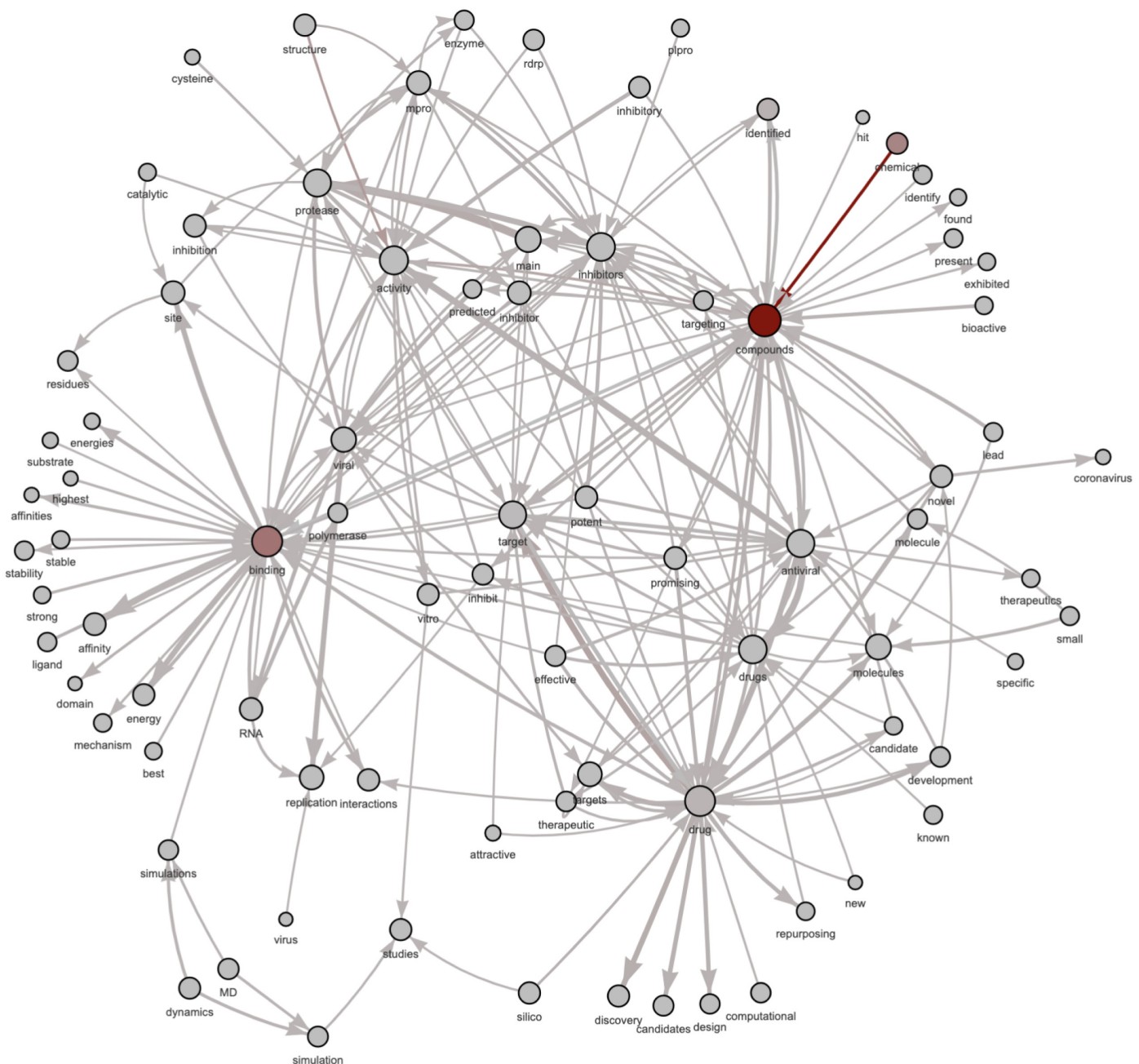

**Fig 12. Filtered network of topic #88.** Nodes are words and an (oriented) edge between two words occurs if they form a bigram. Node size is proportional to the word probability provided by LDA while edge width is proportional to *LDA2Net-weights*. Node and edge color represent betweenness centrality, ranging from gray (*min* observed value) to dark-red (*max* observed value).

of topic #88 are presented (by using a hierarchical network layout). In particular, here, communities are ordered by size. By visual inspection, we can assert that in (a) the community focuses on chemical compounds and antiviral drug design, the subtopic (b) is about `mpro` (abbreviation of main protease) and `inhibitors` and binding mechanisms and (c) concerns viral RNA replication and transcription.

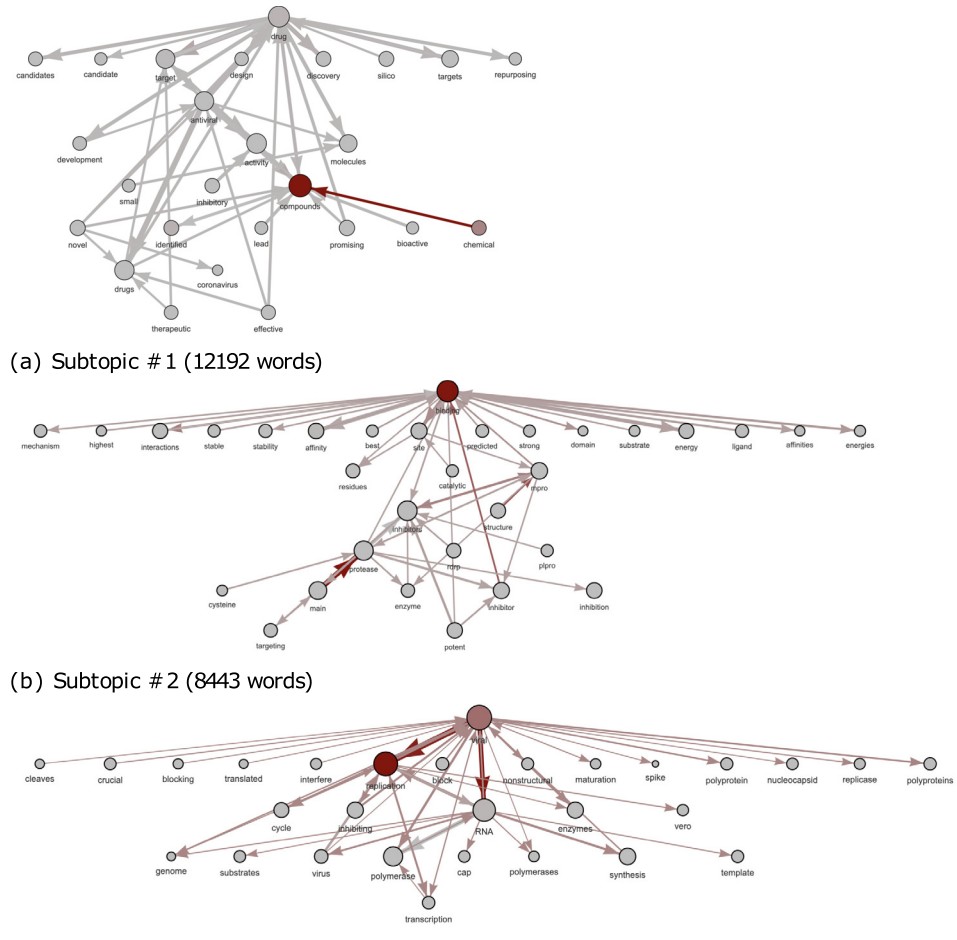

(a) Subtopic # 1 (12192 words)

(b) Subtopic # 2 (8443 words)

(c) Subtopic # 4 (1638 words)

**Fig 13. Top-50-edges graphs of the four largest subtopics for topic #88.** Node size is proportional to the word probability provided by LDA while edge width is proportional to *LDA2Net − weights*. Node and edge color represent betweenness centrality, ranging from gray (*min* observed value) to dark-red (*max* observed value).

However, in this work, we aim to create labels for subtopics without human intervention as well. In other words, once partitioned the network into subtopics, we address to generate label candidates for each subtopic (as well as for the whole topic) automatically. For this reason, we devised an heuristic (described in Automatic Topic Labelling) able to generate short sentences of a few words to be used as subtopic labels. These are sampled phrase fragments of a topic that intend to help understand the type of content. The employed heuristic exploits node and edge metrics to sample random walks of different lengths from topic communities.

Table 2 enumerates, for some subtopics of topic #88, the most frequently sampled *n*-gram label of length 2, 3 and 4. The automatically generated labels seem to capture correctly the contents of subtopics represented in Fig 13. For instance, in the first row, the 3 − *gram* label `antiviral→activity→compounds` expresses the subject of the subtopic nicely, as well as the other labels for the other subtopics. In fact, automatic labels are aligned with human interpretation. This is a crucial point to be stressed: with our method, we can generate subtopic summaries similar to what a human being would do. Even though such a heuristic could be applied to generate label candidates at the topic level, we noticed that in such a case,

the method might produce a vast range of labels, each observed a few times and almost uniformly distributed, clearly an undesired output. Therefore, rather than directly sampling topic labels from topic networks, we suggest using the label candidate of the largest subtopic of a topic as the topic level label.

## Discussion and conclusions

This paper presents a novel approach, called `LDA2Net`, aiming at enriching standard topic modeling algorithms. Specifically, our method, built upon LDA, allows the construction of word networks specific to each topic. By adding direct relations between the LDA topic-specific words, we intend to make the corpus investigation task less tedious and more efficient work. By increasing the transparency and readability of identified topics, this novel framework allows us to explore the architecture of topics in a more user-friendly way. Network visualization and measures and subtopic labels better guide the user through the corpus exploration and consequently improve the user interpretation experience through an effective joint usage of LDA and bigrams information.

In place of simple weighted lists of words provided by LDA, the use of word networks undoubtedly better fits the human interpretation and makes it less arbitrary. Interestingly, it even encodes more knowledge. Indeed, through diverse metrics, some of them based on the network concepts, we were able to assess the improvement brought by `LDA2Net`, in terms of additional information.

Through measures over topic networks, we could identify two classes of topics, structurally different, which likely distinguish cross-cutting topics from specialized ones. Such a distinction can help when it comes to filtering out which topics to focus on first.

Further, the proposed framework favors a better interpretation of the corpus because it easily lets the identification of sub-themes. That is carried out through a community detection approach, which partitions topics into subtopics and then extrapolates key issues at a finer grain. In this regard, the framework offers a technical advantage compared to standard LDA, being more robust to misspecification of the number of topics, thanks to the possibility of identifying subtopics ex-post. In connection with the subtopic identification procedure, we proposed a labeling algorithm based on random walks over the networks. The heuristic generates labels in the form of short sentences of a few words, really close to those a human user would give. This is a promising outcome from the interpretive viewpoint, but it also translates into time-saving. To get a sense of how demanding a labeling task can be, the expert took five hours to give a label to each topic extracted from the Cord-19 dataset (see Table 7 in S1 File), not to mention subtopics.

For the sake of completeness, we applied our approach to other topic modeling methods, such as the Correlated Topic Model (CTM) and Sparse Additive Generative Model (SAGE), (see Appendix F), to highlight the broad applicability of our enrichment technique. There are currently no benchmark techniques for bigram-enriched topic models, so the comparison is only qualitative. Yet, we intend to investigate comparability and benchmark aspects related to the enriched models in future works.

From the point of view of the COVID-19 literature exploration, `LDA2Net` provides a unique, in-depth perspective on critical issues discussed in the scientific literature. In particular, `LDA2Net` facilitates the Cord-19 corpus exploration and understanding process for both specialist and non-specialist audiences, acting as a human-facilitation layer between the LDA outputs and the topic model users. Besides, highly experienced and specialized researchers can gain a broader map of contents outside their domain and thematic cross-connections with

other areas of expertise. Furthermore, search by relevant sets of words can be significantly facilitated by examining topic word networks.

There are numerous natural developments of the `LDA2Net` approach. One could use Part-Of-Speech tags and/or word Dependency Relations as an alternative to bigrams to construct topic-specific relations among words or lemmas and to typify the nodes and relations in topic networks. One could also exploit document-level date-time to create dynamic networks for each topic at the desired granularity or other document-level metadata to generate covariate-value specific topic networks. Moreover, we envisage validating the advantages of `LDA2Net` through a set of experiments with humans, both experts, and non-experts subjects, and using different corpora.

Finally, we plan to publish an open-source `R` library (and possibly also a Python version) to facilitate the deployment, usage and further extension of `LDA2Net`.

## Appendices

## A Topic models

In natural language processing, topic modeling is a widely used technique aiming to automatically find themes (or topics) in text. Topic modeling can be performed by vector space models or Probabilistic Graphical Models (PGMs). PGMs for topic modelling aim at discovering the latent semantic structures of a corpus by relying on a document generation process. The idea behind the document generation process comes from the human written articles. Indeed, when a person writes an article, she or he has some thinking in mind, some topics, and then she or he will extend these themes into some topic related words, which convey the desired meaning. Eventually, these words will be written down to complete an article. Probability topic models simulate the behavior of articles' generating process.

### A.1 Latent Dirichlet Allocation

Latent Dirichlet Allocation (LDA) as defined in [6], is a probabilistic generative model and draws upon this very idea. We define a generative model a machine learning technique generating an output considering the prior distribution of some objects. LDA assigns a distribution of topics to each document, and a distribution of words to each topic to provide low dimensional, probabilistic descriptions of documents and words. In other words, LDA assumes that documents are mixtures of multiple topics, typically not many, and each document is generated by a process. Dirichlet Distributions encode the intuition that documents are related to a few topics. A topic, in turn, is a distribution over a fixed vocabulary and each topic is assumed to be generated first, before the documents. Only the number of topics is specified in advance.

In plain language, the generative process of a document takes two steps. First, a distribution over topics is chosen randomly (that implies a distribution over a distribution for this step). Then, for each word in the document, a topic from the distribution over topics is chosen randomly and next a word from the related topic (distribution over the vocabulary) is picked randomly. Note that words are generated independently of other words (bag-of-words model). Assuming this generative model for a collection of documents, LDA then tries to backtrack from the documents to find a set of topics that are likely to have generated the collection. In practical terms, given a set of documents, and the number of topics we would like to discover out of this set, the output will be the topic model, that is the documents expressed as a combination of the topics. The algorithm provides the weight of connections between documents and topics and between topics and words. Top word lists represent topics for facilitating human interpretation. For a thorough description of LDA we refer the reader to [6].

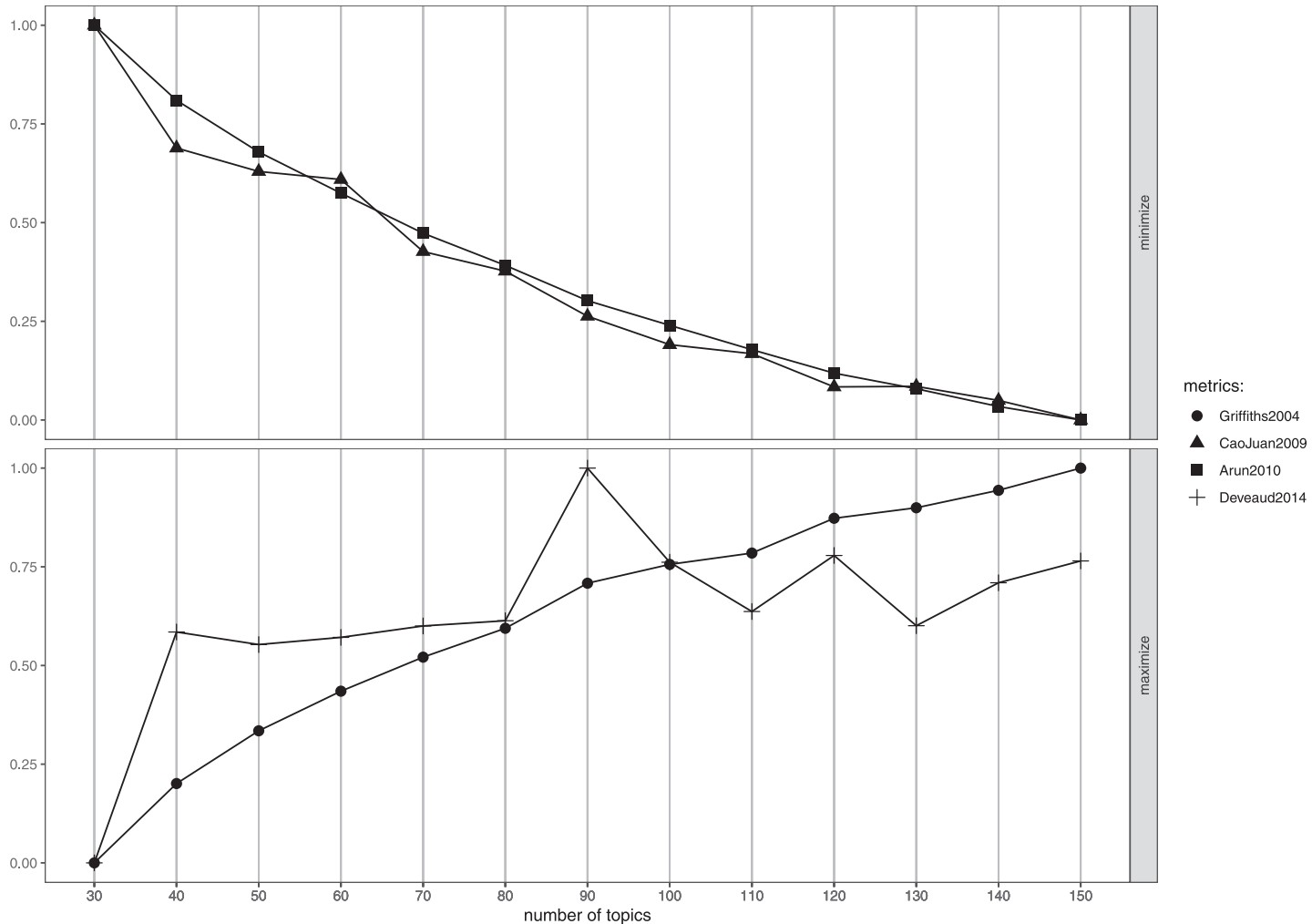

**Fig 14. Average values of the four criteria identified in [24] used for selecting K.**

## A.2 LDA parameters

To select the number of topics *K* for the LDA model, we employed the method proposed by [24] and the R library LDATUNING. For each $K \in$ {30, 40, 50, 60, 70, 80, 90, 100, 110, 120, 130, 140} we estimated multiple LDA models with different seeds and 1300 iterations (with *n. burnin.iter* = 300), using the Gibbs Sampling approach. Then, four criteria [25–28] were employed to choose the optimal value of *K*. Based on the average values of the four considered criteria (see Fig 14), *K* = 120 appeared to be the a good candidate number of topics. Finally, we resumed the best run of the LDA estimation obtained for *K* = 120 running 3000 additional iterations to ensure convergence. The LDA models, and their parameters, have been estimated using the *topicmodels* R library, which interfaces with the C code developed by [6] and the C++ code for fitting LDA models through Gibbs sampling developed by Xuan-Hieu Phan et al. (https://gibbslda.sourceforge.net/). No constraints were imposed on the values for the parameters. For the LDA model with 120 topics, the estimated alpha parameter value is 0.4166667. The beta parameters matrix, of size [34563 x 120], containing the logarithms of word

probabilities for each topic, can be accessed through the following GitHub repository: https://github.com/carlosantagiustina/underthesurfaceofCOVID19topics.

## B Preprocessing

Only words that have more than two characters are considered in this work. Moreover, all abstracts that don't contain at least ten words that occur at least once every one million tokens have been removed. The resulting filtered corpus contains 398, 818 documents (i.e., article abstracts). The average number of terms per document is close to one hundred (98.21) tokens, with a standard deviation of 45.65.

Before tokenizing the corpus, uppercase strings referring to the abstract sections (e.g., "KEY RESULTS") have been removed using a RegEx, as well as arabic and latin numbers. Also, English stopwords from the QUANTEDA [29] R library have been identified through RegEx and removed from the documents' strings. Excluding acronyms, all tokens have been lower-cased. Terms that don't appear at least once every million terms have been removed, and therefore do not belong to the words vocabulary $W$. Remark that, in LDA2Net, tokenization of documents in bigrams takes place once the stopwords have been removed from the documents' strings. Also, during this process, all punctuation characters (except apostrophes and quotation marks) are considered string breaks, which means that two consecutive words that are separated by one or more punctuation characters (different from the aforementioned exceptions) will not be extracted and counted as a bigram. Only the bigrams that are composed by two words (unigrams) contained in in the words vocabulary $W$ are considered. By so doing, bigrams made up by very rare words are also filtered out.

## C Network theory background

### C.1 Network notation

A network is a collection of vertices joined by edges. Vertices and edges are also called nodes and links in computer science. In mathematics, a network is called a graph. A graph $G = (V, E)$ is defined as a set of vertices, $V$, which are connected by a set of edges, $E \in V \times V$, typically represented as a square matrix. Given a graph $G$ with $n$ nodes, the adjacency matrix $A$ of $G$ is a square $n \times n$ matrix. The elements of the adjacency matrix $A = (a_{i,j})$ assume values $a_{i,j} \in \{0, 1\}$, such that

$$A_{i,j} = \begin{cases} 0, & \text{if } (i,j) \notin E \\ 1, & \text{otherwise} \end{cases} \tag{6}$$

that is $a_{i,j} = 1$ if there exists an edge joining nodes $i$ and $j$, and $a_{i,j} = 0$ otherwise. A graph is undirected if edges have no direction. If there is an edge from $i$ to $j$ in an undirected graph, then there is also an edge from $j$ to $i$. This means that the adjacency matrix of an undirected graph is symmetric. A graph is directed if edges have a direction. If there is an edge from $i$ to $j$ in an directed graph, then there is not necessarily an edge from $j$ to $i$, but it might exist. This means that the adjacency matrix of an directed graph is not necessarily symmetric. The adjacency matrix has 0s on the diagonal for simple graphs without self-loops. Many of the networks we study are unweighted, that is, they have edges that form simple on/off connections between vertices. Either they are there or they are not. However, it is helpful to represent edges as having a strength, weight, or value in some situations. The weights in a weighted network are usually positive real numbers, but there is no reason in theory why they should not be negative. These weighted networks can be represented by giving the elements of the adjacency matrix values equal to the weights of the corresponding connections.

## C.2 Network measures

In order to characterize the structure of a graph, it is crucial to study and quantify its properties. These properties can be organized into three levels of abstraction:

1. *element-level*: to identify the most important nodes/links of the network

2. *group-level*: to find cohesive groups of nodes in the network

3. *network-level*: to study topological properties of networks as a whole

**C.2.1 Element-level analysis.** In network analysis, *element-level* properties are used to measure the level of importance of a single component of a graph with respect to the others. Element-level descriptors, also called *centrality measures*, are a crucial tool for understanding networks. These topological indicators, adopted to score both nodes and edges, are scalar values assigned to each node (edge) in the graph in order to quantify the node's (edge's) importance based on a certain assumption. In contrast, network-level measurements, calculated over the whole network, provide overall indications about the network structure. For our analysis, we take into account the following centralities:

*Degree centrality* (node). The degree centrality is the degree of a vertex, the number of edges connected to it. In directed networks, vertices have both an in-degree and an out-degree, and both may be useful as measures of centrality in the appropriate circumstances.

*Betweenness centrality* (node). The betweenness centrality [30] over nodes measures the extent to which a vertex lies on paths between other vertices. This centrality detects the amount of influence a node has over the flow of information in a graph and thus it identifies nodes that serve as a bridge from one part of a graph to another. It is a measure based on shortest paths: for each vertex it is equal to the number of shortest (geodesic) paths that pass through the vertex.

*PageRank* (node). The *PageRank* centrality [31] is the trade name given it by the Google web search corporation, which has adopted it as key part of the web ranking technology. It is particularly suitable for directed network as it accounts for link direction. Each node in a network is assigned a score based on its indegree. These links are also weighted based on the relative score of its originating vertex. In this way nodes with many incoming links are influential, and nodes to which they are connected share some of that influence. *PageRank* can help uncover influential nodes whose reach extends beyond just their direct connections.

*Betweenness centrality* (edge). The betweenness centrality over edges [32] is the sum of the fraction of all-pairs shortest paths that pass through that edge. Usually, important bridge-like connectors between two parts of a network have high betweenness centrality as they have a significant influence on the transfer of information through the network.

**C.2.2 Group-level analysis.** In the social context, a community is a set of individuals interacting with each other more frequently than with those outside the group. In this regard *community detection* is discovering groups where individuals' group memberships are not explicitly given. Similarly, in network analysis, a community is a group of nodes, which are highly connected to each other than to the rest of the nodes in the network [33]. In the last years, community structure has increasingly become the most-studied structural feature of complex networks. The approach consists in dividing a network into subgroups by grouping nodes that are tightly coupled to each other and loosely coupled to the rest of the vertices in the network [34]. However, gauging the intuitive concept of community structure is not trivial. One of the most effective approaches to this issue has been defining a quality function able to estimate the strength of division of a network into modules [34, 35]. This strength has been

quantified by using the measure known as *modularity* [36], computed as the fraction of the edges that fall within the given groups minus the expected value of the fraction if edges were distributed at random (for more details we refer the reader to [37]). Although modularity optimization is an NP-hard problem, many community detection algorithms are based on this principle. The most famous one is probably the Louvain algorithm [38]. Yet, many other approaches exist to identify communities, for instance, by simulating random walks inside the network. The general idea is that, given a graph and a starting point, if we select a neighbour at random and move to the selected neighbour and repeat the same process till a termination condition, this walk, namely a random sequence of nodes, is more likely to stay within the same community as there are only a few edges that lead outside a given community. In other words, a *random walker* will tend to wander inside densely connected areas of the graph. This concepts inspired the Walktrap algorithm [23] and the Infomap algorithm [39]. Walktrap uses short random walks (of 3-4-5 steps) to compute distances between nodes. Nodes are assigned to groups via bottom-up hierarchical clustering based on (small) intra and (larger) inter-community distances. In this respect, modularity score can be used to select where to cut the dendrogram. It should be pointed out that this algorithm considers only one community per node, which in some cases can be an incorrect hypothesis. For our experiments we use the method in [23]. We employed IGRAPH's weighted walktrap community algorithm implementation, using 4-steps random walks based on *LDA2Net*-weights.

**C.2.3 Network-level analysis.** A classic property shared by many real networks is clustering, also called transitivity. It measures the probability that the adjacent vertices of a vertex are connected. In other words, the clustering coefficient (CC) measures the degree to which nodes in a graph tend to cluster together. There exist two versions of it: a global version designed to provide an overall indication of the clustering in the network and a local version that gives an idea of the embeddedness of single nodes. In its global formulation, the clustering coefficient considers triplets of nodes. By triplet we mean three nodes that are connected by either two (open triplet) or three (closed triplet) undirected ties (see Fig 15); here, a triangle is three closed triplets, one centered on each of the nodes. The global clustering coefficient is the number of closed triplets (or 3 × triangles) over the total number of triplets (both open and closed). A generalization of the global clustering coefficient to weighted networks (networks with weighted edges) was proposed by [21]. The authors first propose a novel definition of the local clustering coefficient—measuring the local cohesiveness considering both the importance of the clustered structure and the interaction intensity (e.g., weights) found on the local triplets—and then the global version, the weighted clustering coefficient averaged over all vertices.

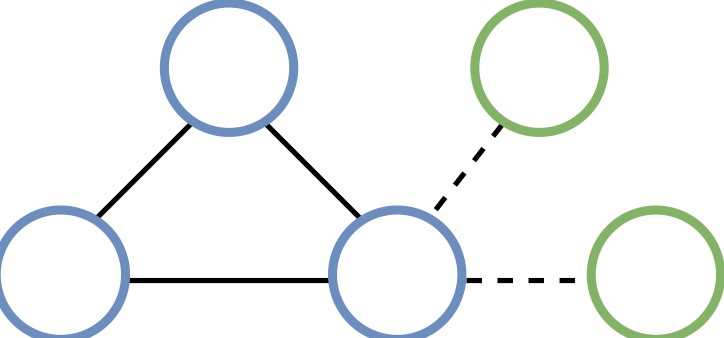

**Fig 15. An example of closed and open triplets.**

## D Jensen-Shannon divergence

The Jensen-Shannon divergence (JSD) is a measure assessing the difference between probability distributions, that is the ground truth and the simulation, by calculating the mutual information between two probability distributions, quantified by gauging the difference between entropies associated with those probability distributions. The JSD is a symmetrized and smoothed version of the Kullback-Liebler divergence. In other words, it is a mutual information measure for assessing the similarity between two probability distributions. It is defined as

$$D_{JS}(P, Q) = \frac{1}{2} D_{KL}(P\|M) + \frac{1}{2} D_{KL}(M\|Q) \tag{7}$$

where $D_{KL}$ is the classical Kullback-Leibler divergence and $M = \frac{P+Q}{2}$.

The generalization of the Jensen-Shannon divergence, in case of more than two probability distributions, is based on the Shannon entropy as follows

$$\begin{aligned} D_{JS_{\pi_1,\ldots,\pi_n}}(P_1, P_2, \ldots, P_n) &= \sum_i \pi_i D(P_i\|M) \\ &= H\left(\sum_{i=1}^{n} \pi_i P_i\right) - \sum_{i=1}^{n} \pi_i H(P_i) \end{aligned}$$

where $\pi_1, \ldots, \pi_n$ are the weights of the probability distributions, $M = \sum_{i=1}^{n} \pi_i P_i$ and $H(P)$ is the Shannon entropy for distribution $P$.

When using the base 2 logarithm, the Jensen–Shannon divergence for two probability distributions ranges between 0 and 1 [40].

## E Gaussian finite mixture model

Automatic model (VEV: ellipsoidal & equal volume) and number of clusters selection (3) based on BIC criterion. See MCLUST R library [41].

All measures were standardised (to have zero means and unitary SEs) before the clustering procedure. Inferred cluster-specific mean and SE are displayed in black.

## F Comparison between models

We compared and applied our approach to three topic modeling methods: LDA, Correlated Topic Model (CTM), and Sparse Additive Generative Models (SAGE). Model implementation parameters are explained in the next section. We employed the same mechanism for both methods, that is, we extracted the bigrams, we performed the topic modeling and then by using the output matrices and the bigram weights we built the graphs for each topic.

In order to compare similar topics, we computed the Jaccard Similarity between the top 25 words of each topic. In particular, we selected for the CTM and SAGE models the topics that are closer to topics #50 and #88 obtained with LDA. Results are reported in Table 5. The fact that the LDA topic #50 is close to topic #95 for both SAGE and CTM is no error: it is due to the Spectral initialization technique, which was used for both models. We computed the Jensen-Shannon Divergence between edge weights as well (see Table 6). As expected, the

**Table 5. Jaccard similarity between topics of different methods.**

|         | topic | SAGE | CTM  |
|---------|-------|------|------|
| LDA #50 | 95    | 0.36 | 0.56 |
| LDA #88 | 72    | 0.49 | 0.56 |

**Table 6. Jensen-Shannon divergence (JS) between topics of different methods.**

|  | topic | SAGE | CTM |
|---|---|---|---|
| LDA #50 | 95 | 0.83137 | 0.83244 |
| LDA #88 | 72 | 0.83216 | 0.83250 |

divergence values are very high. That will be confirmed later in the difference between the graphs obtained by different methods. Once found the corresponding topic, to visually inspect and compare topics (see Fig 16 for wordcloud comparison), we selected the top 30 edges for each network. That means we plotted only those nodes (words) belonging to the edge set (given by top 30 bigrams). Comparison are shown in Figs 17 and 18. Interestingly, even if they have a very similar initial node set, the final networks are different as paths built by top edges are not the same.

While the networks of enriched CTM and SAGE models are akin in some respects, an outcome we anticipated due to their common initialization, LDA appears so different from the other two that any qualitative comparison of the content becomes complex. We defer to future work for the development of more advanced benchmarking and comparison techniques.

## F.1 Estimation and convergence parameters

While the LDA model's parameters have already been discussed in Appendix A.2, we here report details about CTM and SAGE implementation.

The input matrices used for the CTM and SAGE models are identical to the ones used for the LDA model presented in the sections above. As for the LDA model, for both CTM and SAGE we set $K = 120$, to facilitate comparability between models. Also, to facilitate comparability between models, we initialized both models using the Spectral initialization technique from the STM R library, as proposed by Lee and [42]. Spectral initialization parameters are all set to their default values. This technique allows model outcomes to be deterministic, and hence more easily reproducible.

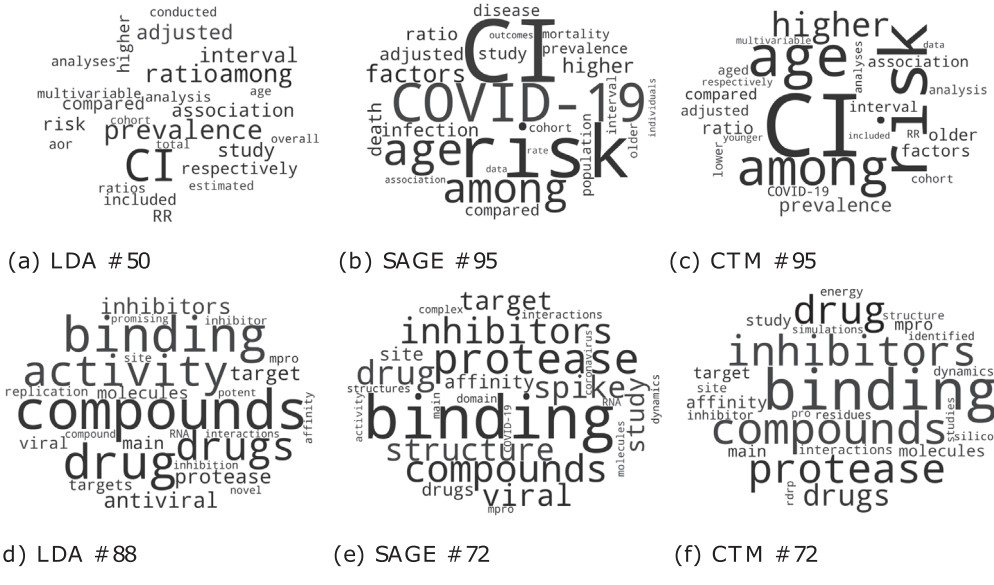

(a) LDA #50    (b) SAGE #95    (c) CTM #95

(d) LDA #88    (e) SAGE #72    (f) CTM #72

**Fig 16. Wordcloud comparison.**

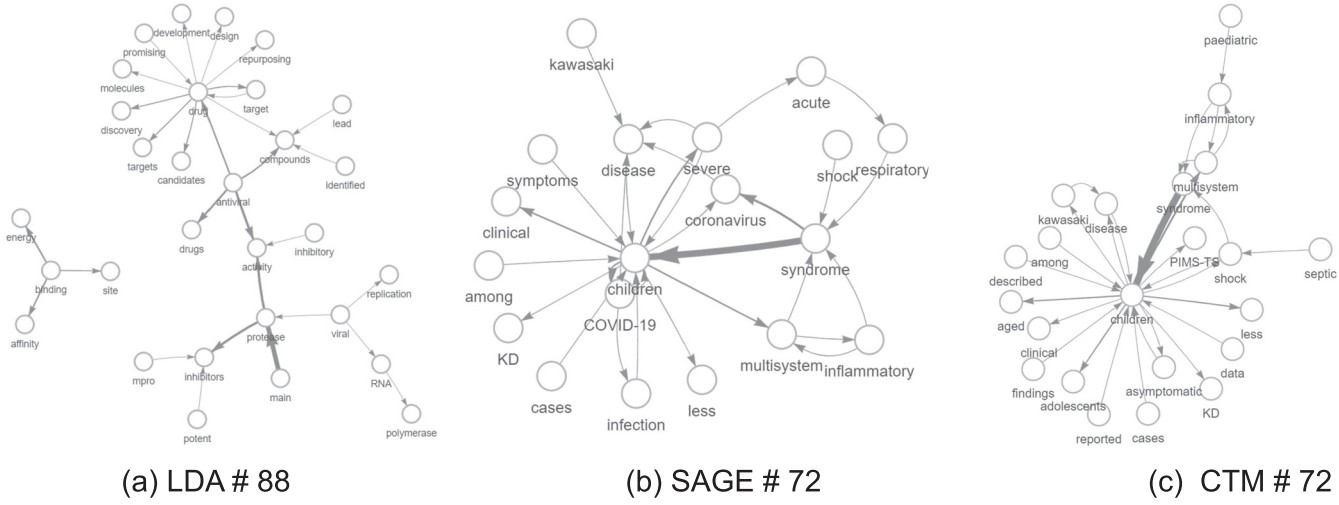

(a) LDA # 88          (b) SAGE # 72          (c)  CTM # 72

**Fig 17. Topic comparison: LDA #88 and SAGE, CTM #72.**

For both the CTM [8] and the SAGE [43] model we use the variational expectation-maximization (VEM) algorithm implemented with the STM library [10], we set all parameters to their default values, except for the maximum number of expectation-maximization iterations (max.em.its), that we set to 100. Since both models converge before reaching the aforementioned value, this parameter choice is irrelevant.

The CTM converged after 65 iterations (Lower Bound: -286094170). The SAGE model converged after 5 iterations (Lower Bound: -291762273).

Using the methods proposed by [10] we checked the Beta matrix and the residuals of the CTM and SAGE model and no anomalies / errors have been detected.

Since we are interested in visually comparing how different enriched models (LDA vs CTM vs SAGE) compare to each other, we apply to the outputs of both newly estimated models (CTM and SAGE) the bigrams-enrichment technique previously implemented with LDA. We

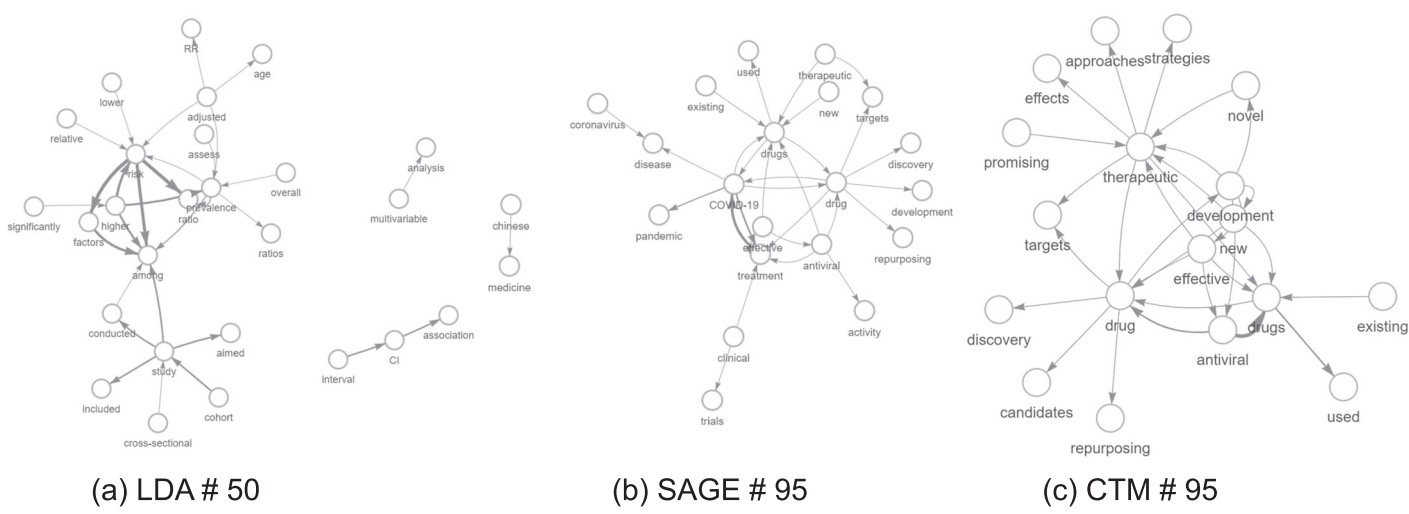

(a) LDA # 50          (b) SAGE # 95          (c) CTM # 95

**Fig 18. Topic comparison: LDA #50 and SAGE, CTM #95.**

recall that this enrichment technique is deterministic and independent from the model used for estimating the words-by-topic (TxW) distributions contained in the **M** matrix.

## Supporting information

**S1 File.**
(PDF)

## Acknowledgments

We thank Dr. Martina Gerotto for her expert support in topic labelling.

## Author Contributions

**Conceptualization:** Giorgia Minello, Carlo Romano Marcello Alessandro Santagiustina, Massimo Warglien.

**Data curation:** Carlo Romano Marcello Alessandro Santagiustina.

**Methodology:** Giorgia Minello, Carlo Romano Marcello Alessandro Santagiustina, Massimo Warglien.

**Writing – original draft:** Giorgia Minello, Carlo Romano Marcello Alessandro Santagiustina, Massimo Warglien.

**Writing – review & editing:** Giorgia Minello, Carlo Romano Marcello Alessandro Santagiustina, Massimo Warglien.

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
