## [Decision Letter · Decision Letter 0]

12 Jun 2023

PONE-D-23-09182LDA2Net

Digging Under the Surface of COVID-19 scientific literature Topics via a Network-Based ApproachPLOS ONE

Dear Dr. Minello,

Thank you for submitting your manuscript to PLOS ONE. After careful consideration, we feel that it has merit but does not fully meet PLOS ONE’s publication criteria as it currently stands. Therefore, we invite you to submit a revised version of the manuscript that addresses the points raised during the review process.

We look forward to receiving your revised manuscript.

Kind regards,

Fu Lee Wang

Academic Editor

PLOS ONE

“C.S. and M.W. acknowledge financial support from the European Union Horizon

2020 project ISEED (Grant Agreement No.

960366).

C.S. acknowledges financial support

from the European Union Horizon 2020

project MUHAI (Grant Agreement No.

951846).

C.S. acknowledges financial support from PON R&I 2014-2020 (FSE REACT-EU)”

“We thank Dr. Martina Gerotto for her 743

expert support in topic labelling. 744

C.S. and M.W. acknowledge financial 745

support from the European Union Horizon 746

2020 project ISEED (Grant Agreement No. 747

960366). 748

C.S. acknowledges financial support 749

from the European Union Horizon 2020 750

project MUHAI (Grant Agreement No. 751

951846).”

“C.S. and M.W. acknowledge financial support from the European Union Horizon

2020 project ISEED (Grant Agreement No.

960366).

C.S. acknowledges financial support

from the European Union Horizon 2020

project MUHAI (Grant Agreement No.

951846).

C.S. acknowledges financial support from PON R&I 2014-2020 (FSE REACT-EU)”

Reviewers' comments:

Reviewer's Responses to Questions

**Comments to the Author**

1. Is the manuscript technically sound, and do the data support the conclusions?

Reviewer #1: Partly

Reviewer #2: Yes

2. Has the statistical analysis been performed appropriately and rigorously? 

Reviewer #1: No

Reviewer #2: No

3. Have the authors made all data underlying the findings in their manuscript fully available?

Reviewer #1: Yes

Reviewer #2: No

4. Is the manuscript presented in an intelligible fashion and written in standard English?

Reviewer #1: Yes

Reviewer #2: Yes

5. Review Comments to the Author

Reviewer #1: 1) Your references are out-of-date. You should cite up-to-date studies.

• NET-LDA: a novel topic modeling method based on semantic document similarity

• Concept-LDA: Incorporating Babelfy into LDA for aspect extraction

• Combining semantic graph and probabilistic topic models for discovering coherent topics

• …

2) What is your LDA parameter values? For example, alpha, beta., …. There is no information about realization of LDA.

3) You evaluate obtained results in many ways. But beside these, you should evaluate your extracted topics in terms of topic coherence, tf-idf coherence, f-score, precision, recall. You should compare your proposed topic model with LDA, STM, CTM.

Reviewer #2: The manuscript investigates Covid-19-related scientific literature using topic modeling techniques. This work proposes LDA2Net, a deep learning-based LDA model, and exploits the frequencies of consecutive words pairs to conduct topic analysis. The paper is generally well-written and technically sound. However, the manuscript requires further enhancement on the following points:

1. The manuscript does not provide additional insightful analysis apart from the topic analysis, which significantly restricts the contribution of the work. The topic analysis has some degree of randomness due to the initialization, and the outputs are based on statistics. Therefore, the manual interpretation and identification of the topic meanings are essential. Viewing the specific meaning of the target topic, Covid-19 scientific literature, the missing part is very important to distinguish the manuscript from some experiment reports.

2. The preprocessing method is questionable. The filtering method only removes the articles uploaded before December 31, 2019. Is it a reasonable method? Although the rest of papers are concerning SARS and MERS, are they all related to Covid-19? The inclusive and exclusive criteria of paper filtering are necessary to enhance the trustworthiness of the paper. You may reference the following papers which focus on literature surveying:

Information fusion and artificial intelligence for smart healthcare: a bibliometric study

A bibliometric review of soft computing for recommender systems and sentiment analysis

3. There is one most concerned point: is it correct to claim the paper proposes an automatic topic labeling? Topic labeling should be the identification and interpretation of the identified topics (word clusters). In my understanding, the paper only discovers the word clusters, but does not provide an automatic method for topic meaning identification. The authors are required to justify this point or revise the manuscript.

4. I am also curious about the mapping and agreement between the identified topics and the 120 topics in the dataset (again, are all the 120 topics concerning Covid-19?).

6. PLOS authors have the option to publish the peer review history of their article (what does this mean?). If published, this will include your full peer review and any attached files.

Reviewer #1: No

Reviewer #2: No

---

## [Author Response · Author response to Decision Letter 0]

21 Sep 2023

Please see the file Response to Reviewers containing the response to both reviewers and the editor.

---

## [Decision Letter · Decision Letter 1]

10 Oct 2023

PONE-D-23-09182R1LDA2Net

Digging Under the Surface of COVID-19 scientific literature Topics via a Network-Based ApproachPLOS ONE

Dear Dr. Minello,

Thank you for submitting your manuscript to PLOS ONE. After careful consideration, we feel that it has merit but does not fully meet PLOS ONE’s publication criteria as it currently stands. Therefore, we invite you to submit a revised version of the manuscript that addresses the points raised during the review process.

Please conduct a comparison study between your model and other models in your revised version.  Otherwise, it is difficult to judge the technical merit of your study.

We look forward to receiving your revised manuscript.

Kind regards,

Fu Lee Wang

Academic Editor

PLOS ONE

Reviewers' comments:

Reviewer's Responses to Questions

**Comments to the Author**

1. If the authors have adequately addressed your comments raised in a previous round of review and you feel that this manuscript is now acceptable for publication, you may indicate that here to bypass the “Comments to the Author” section, enter your conflict of interest statement in the “Confidential to Editor” section, and submit your "Accept" recommendation.

Reviewer #1: All comments have been addressed

Reviewer #2: All comments have been addressed

2. Is the manuscript technically sound, and do the data support the conclusions?

Reviewer #1: Partly

Reviewer #2: Yes

3. Has the statistical analysis been performed appropriately and rigorously? 

Reviewer #1: No

Reviewer #2: Yes

4. Have the authors made all data underlying the findings in their manuscript fully available?

Reviewer #1: Yes

Reviewer #2: No

5. Is the manuscript presented in an intelligible fashion and written in standard English?

Reviewer #1: Yes

Reviewer #2: Yes

6. Review Comments to the Author

Reviewer #1: I understand your aim but the comparsion between models was needed in this study. You should compare your proposed topic model with LDA, STM, CTM or any other models.

Reviewer #2: The authors' responses have covered most of my concerns. The paper could be considered for publication after minor revisions.

My concern is about Fig. 1. From the diagram, the graph contains both directional and undirectional edges. But I did not find the contents mentioning the edges' property. Moreover, it will be needed to show how to obtain the graph based on T_{n}, and how to determine if the edge is directed or undirected.

7. PLOS authors have the option to publish the peer review history of their article (what does this mean?). If published, this will include your full peer review and any attached files.

Reviewer #1: No

Reviewer #2: No

---

## [Author Response · Author response to Decision Letter 1]

3 Jan 2024

Please see Response to Reviewers file

---

## [Decision Letter · Decision Letter 2]

23 Feb 2024

LDA2Net

Digging Under the Surface of COVID-19 scientific literature Topics via a Network-Based Approach

PONE-D-23-09182R2

Dear Dr. Minello,

We’re pleased to inform you that your manuscript has been judged scientifically suitable for publication and will be formally accepted for publication once it meets all outstanding technical requirements.

Kind regards,

Fu Lee Wang

Academic Editor

PLOS ONE

Additional Editor Comments (optional):

Reviewers' comments:

Reviewer's Responses to Questions

**Comments to the Author**

1. If the authors have adequately addressed your comments raised in a previous round of review and you feel that this manuscript is now acceptable for publication, you may indicate that here to bypass the “Comments to the Author” section, enter your conflict of interest statement in the “Confidential to Editor” section, and submit your "Accept" recommendation.

Reviewer #1: All comments have been addressed

Reviewer #2: All comments have been addressed

2. Is the manuscript technically sound, and do the data support the conclusions?

Reviewer #1: Yes

Reviewer #2: Yes

3. Has the statistical analysis been performed appropriately and rigorously? 

Reviewer #1: Yes

Reviewer #2: Yes

4. Have the authors made all data underlying the findings in their manuscript fully available?

Reviewer #1: Yes

Reviewer #2: Yes

5. Is the manuscript presented in an intelligible fashion and written in standard English?

Reviewer #1: Yes

Reviewer #2: Yes

6. Review Comments to the Author

Reviewer #1: You take into account my all recommendations and improve paper based on them. So, I accept your paper.

Reviewer #2: The authors have addressed my comments. I recommend the journal to accept and publish the manuscript.

7. PLOS authors have the option to publish the peer review history of their article (what does this mean?). If published, this will include your full peer review and any attached files.

Reviewer #1: No

Reviewer #2: No

---

## [Editor Report · Acceptance letter]

21 Mar 2024

PONE-D-23-09182R2 

PLOS ONE

Dear Dr. Minello, 

I'm pleased to inform you that your manuscript has been deemed suitable for publication in PLOS ONE. Congratulations! Your manuscript is now being handed over to our production team.

Kind regards, 

on behalf of

Professor Fu Lee Wang 

Academic Editor

PLOS ONE